# VFScale: Intrinsic Reasoning through Verifier-Free Test-time Scalable Diffusion Model

**Tao Zhang**[1,2*] **Jia-Shu Pan**[2*] **Ruiqi Feng**[2] **Tailin Wu**[2*†]
[1]Zhejiang University
[2]Department of Artificial Intelligence, School of Engineering, Westlake University
`{zhangtao,panjiashu,wutailin}@westlake.edu.cn`

## Abstract

Inspired by human SYSTEM 2 thinking, LLMs excel at complex reasoning tasks via extended Chain-of-Thought. However, similar test-time scaling for diffusion models to tackle complex reasoning remains largely unexplored. From existing work, two primary challenges emerge in this setting: (i) the dependence on an external verifier indicating a notable gap from intrinsic reasoning of human intelligence without any external feedback, and (ii) the lack of an efficient search algorithm. In this paper, we introduce the Verifier-free Test-time Scalable Diffusion Model (VFScale) to achieve scalable intrinsic reasoning, which equips number-of-sample test-time scaling with the intrinsic energy function of diffusion models as the verifier. Concretely, VFScale comprises two key innovations to address the aforementioned challenges. On the training side, VFScale consists of a novel MRNCL loss and a KL regularization to improve the energy landscape, ensuring that the learned energy function itself serves as a reliable verifier. On the inference side, VFScale integrates the denoising process with a novel hybrid Monte Carlo Tree Search (hMCTS) to improve search efficiency. On challenging reasoning tasks of Maze and Sudoku, we demonstrate the effectiveness of VFScale's training objective and scalable inference method. In particular, trained with Maze sizes of up to $6 \times 6$, our VFScale solves 88% of Maze problems with much larger sizes of $15 \times 15$, while standard diffusion models completely fail. The code can be found at https://github.com/AI4Science-WestlakeU/VFScale.

## 1 Introduction

Human intelligence can allocate more computation to solve harder problems, known as SYSTEM 2 thinking (Kahneman, 2011). Motivated by this, large language models (LLMs) based on autoregressive models have exhibited promising performance in reasoning tasks through deliberate long Chain-of-Thought (Lightman et al., 2023; DeepSeek-AI et al., 2025). Alongside autoregressive models, diffusion models have recently emerged as a compelling alternative. By casting reasoning as an optimization problem, these models iteratively refine candidate solutions toward higher-quality outputs. This approach has demonstrated strong potential, achieving notable results on tasks such as Sudoku solving, graph connectivity, and shortest-path computation (Du et al., 2022; 2024).

Although the iterative denoising procedure of diffusion models mirrors the repeated reasoning cycles required by complex reasoning tasks, their performance declines sharply once the problem difficulty exceeds the training distribution (Du et al., 2022). Moreover, prior works show that simply increasing the number of sampling steps quickly leads to a performance plateau (Du et al., 2024). Instead, in image generation, inference-time scaling to improve performance by increasing the number of samples has proven effective (Ma et al., 2025). This success, however, heavily relies on a dense external verifier (Ouyang et al., 2022; Lee et al., 2024), which provides continuous guidance by scoring sample quality. Moreover, these external verifiers are often learned from extensive labeled

---

*Equal contribution.
†Corresponding author.

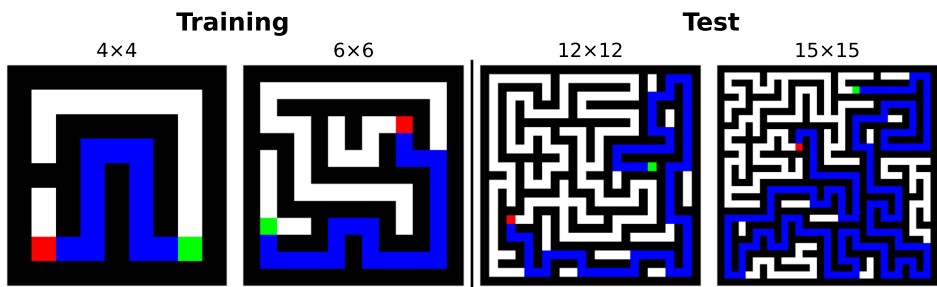

Figure 1: Visualizations of Maze training data and solutions generated by hMCTS denoising of our VFScale framework.

datasets that are particularly difficult to obtain for reasoning tasks. In contrast, human intelligence can engage in deep reflective intrinsic reasoning without external feedback, revealing a clear gap with such methods (Lombrozo, 2024; Chi et al., 1994; Huang et al., 2024). These observations motivate our central question: *Can we design a Verifier-free Test-time Scalable Diffusion Model to achieve scalable intrinsic reasoning?*

In pursuit of such **a test-time scalable intrinsic reasoning model**, we leverage the fundamental mechanics of diffusion models. Diffusion models approximate the score function (Song et al., 2021b), which is equivalent to the negative gradient of the energy function. Our key insight is that the energy function, as a measure of the learned probability, can serve as an effective verifier itself. Based on the above premise, in this work, we explore equipping **number-of-sample test-time scaling** with diffusion models' **intrinsic energy function**.

However, this strategy presents two primary challenges: **(1)** Firstly, using number-of-sample test-time scaling requires a verifier to evaluate and select samples. Prior work on test-time scaling has also established that verifier accuracy is crucial for performance (Setlur et al., 2025; Ma et al., 2025). Consequently, using the energy function as an intrinsic verifier requires it to reliably reflect the sample quality. Nevertheless, previous energy-based diffusion models have largely ignored this requirement. **(2)** Secondly, the efficiency of the search algorithm is critical when scaling by varying the number of samples (Snell et al., 2025). However, the search efficiency of existing methods, such as best-of-$N$ (BoN), warrants further enhancement.

To approach these challenges, we propose Verifier-Free Test-Time Scalable Diffusion Models (VFScale), a novel framework for solving complex reasoning problems. VFScale consists of innovations in both training objectives and inference scaling as shown in Fig. 2 to respectively address the two aforementioned challenges. **(1) On the training side,** VFScale incorporates a novel Monotonic-Regression Negative Contrastive Learning (MRNCL) objective and a KL regularization to improve the energy landscape, especially to better align the energy function with sample quality with MRNCL loss. The MRNCL objective samples two negatives[1] at different corruption levels for each positive sample, and requires their energy values to reflect their L2 distances to the positive. We refer to this alignment as *performance-energy consistency*.[2] The KL regularization term backpropagates through the denoising process, smoothing the energy landscape. **(2) On the inference side,** we propose a novel hybrid Monte Carlo Tree Search (hMCTS) that strategically balances exploration and exploitation during the denoising process. In the noisy initial stages, it employs Best-of-N (BoN) for broad, parallel exploration, preventing the premature elimination of promising solution pathways. As denoising progresses and the search space narrows, it transitions to MCTS for deep, fine-grained exploitation of the most viable candidates. This hybrid strategy enables VFScale to efficiently translate increased test-time computation into substantial gains in reasoning performance.

We demonstrate the effectiveness of our VFScale's training objective and scalable inference method on challenging reasoning tasks of Maze and Sudoku. In Sudoku tasks, our VFScale solves $43\%$ problems when conditioned on much **less given digits (out-of-distribution condition)**, while the standard diffusion model only solves $30\%$ problems. In Maze tasks, trained with Maze sizes of up

---

[1]Here, positive samples refer to data points, while negative samples correspond to noise-perturbed versions of the data.

[2]Detailed description for *performance-energy consistency* can be found in Appendix A.

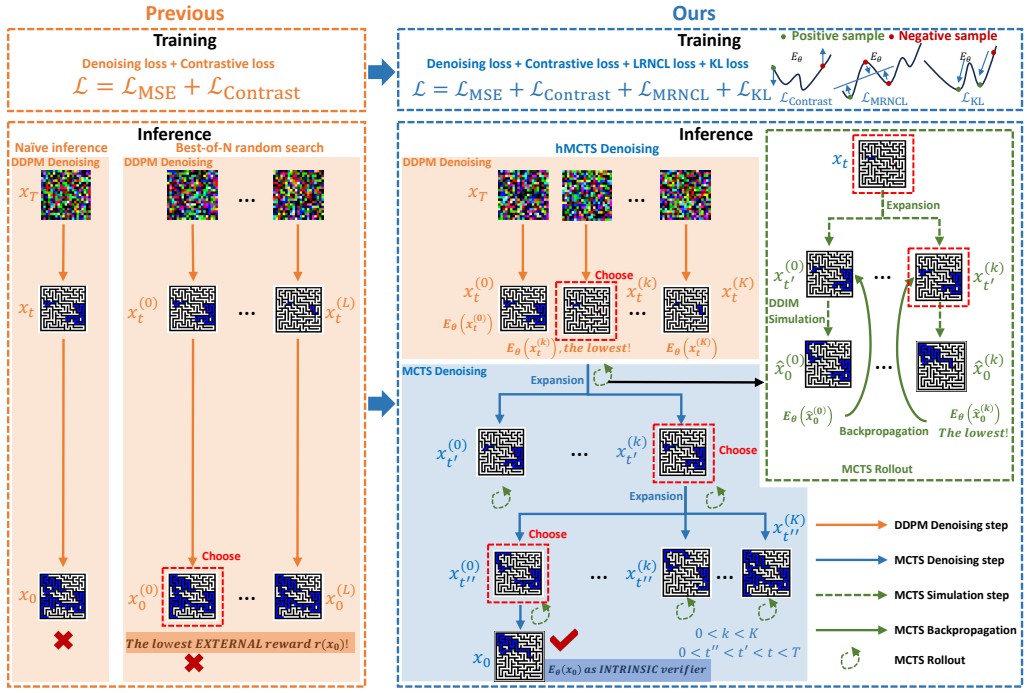

Figure 2: Overview of VFScale. This figure illustrates the key aspects of VFScale by contrasting its training and inference strategies with those of the previous method. **(1)** To qualify the intrinsic energy of diffusion models as a verifier, VFScale introduces $\mathcal{L}_{\text{MRNCL}}$ and $\mathcal{L}_{\text{KL}}$ to improve the energy landscape during training. **(2)** In order for a higher search efficiency, VFScale proposes hybrid Monte Carlo Tree Search (hMCTS) that achieves a balance between best-of-$N$ and MCTS.

to $6 \times 6$, VFScale successfully solves $88\%$ of substantially larger $15 \times 15$ instances (see Fig. 1), whereas the standard diffusion model fails entirely.

In summary, our contributions are as follows: **(1)** We introduce Verifier-free Test-time Scalable Diffusion Models (VFScale), a novel framework that can scale up test-time computation for better reasoning capability. **(2)** We introduce Monotonic-Regression Negative Contrastive Learning (MRNCL) and KL regularization into the training objective to improve the energy landscape. **(3)** We integrate a hybrid Monte Carlo Tree Search into the denoising process, enabling test-time scaling with better search efficiency. **(4)** We demonstrate the effectiveness of our method on challenging Sudoku and Maze datasets **where the conditions are out-of-distribution or the problem sizes are much larger.**

## 2 RELATED WORK

**Diffusion for Reasoning**: The iterative refinement process inherent in diffusion models lends itself well to tasks requiring multistep reasoning. Du et al. (2022) firstly formulated multi-step reasoning as an energy minimization problem. To approach harder problems with more complex energy landscapes, more compute budget is used for optimization steps. Du et al. (2024) formulated the energy-based model as a diffusion model, and the performance on harder tasks was significantly improved. While effective, the performance gain by scaling the compute of these methods quickly saturates as the number of sampling steps increases. Building upon Du et al. (2024), VFScale instead explores verifier-free number-of-sample scaling and proposes corresponding training and inference strategies.

**Test-time Scaling of Diffusion**: Improving diffusion model samples by increasing compute resources at inference time has emerged as an increasingly important direction. Early approaches focused on increasing the number of sampling steps, leading to diminishing returns (Ho et al., 2020; Karras et al., 2022). More recent work (Ma et al., 2025; Yoon et al., 2025) has explored alternative scaling

dimensions, such as increasing the number of sampled candidates, though these methods typically rely on external verifiers to guide solution selection. VFScale advances beyond these methods by leveraging the intrinsic energy function to guide test-time search, a capability further cultivated through customized training and inference strategies. These innovations make VFScale a versatile and scalable solution for real-world reasoning tasks.

## 3 PRELIMINARY

The Denoising Diffusion Probabilistic Model (DDPM) (Ho et al., 2020) contains a predefined forward process to corrupt data into Gaussian noises, and a learnable reverse process to generate new samples from them. The forward process follows a Gaussian transition kernel $q(\boldsymbol{x}_t|\boldsymbol{x}_{t-1}) = \mathcal{N}(\boldsymbol{x}_t|\sqrt{\alpha_t}\boldsymbol{x}_{t-1}, (1-\alpha_t)\boldsymbol{I}), t = 1, \cdots, T$, where the noise schedule $\{\alpha_t\}_{t=1}^T$ and $T$ is set to make $\boldsymbol{x}_T$ follow approximately a standard Gaussian distribution. The reverse process can be learned to predict the noise from the corrupted data by minimizing

$$\mathcal{L}_{\text{MSE}} = \mathbb{E}_{\boldsymbol{x}_0,\boldsymbol{\epsilon},t}\left[\left\|\boldsymbol{\epsilon} - \boldsymbol{\epsilon}_\theta\left(\sqrt{\bar{\alpha}_t}\boldsymbol{x}_0 + \sqrt{1-\bar{\alpha}_t}\boldsymbol{\epsilon}, t\right)\right\|_2^2\right], \tag{1}$$

where the expectation is w.r.t. $\boldsymbol{x}_0 \sim p(\boldsymbol{x})$, $\boldsymbol{\epsilon} \sim \mathcal{N}(\boldsymbol{0}, \boldsymbol{I})$, and $t \sim \{1, ...T\}$. $\bar{\alpha}_t := \prod_{i=1}^t \alpha_i$. Eq. 1 is equivalent to optimizing a reweighted variational bound on negative log-likelihood. Without loss of generality, in this work, we parameterize $\boldsymbol{\epsilon}_\theta$ as the negative gradient of the energy function $\nabla_{\boldsymbol{x}_t} E_\theta(\boldsymbol{x}_t, t)$ as in Du et al. (2023). To improve the energy landscape, Du et al. (2024) introduced a contrastive loss

$$\mathcal{L}_{\text{Contrast}} = \mathbb{E}_{\boldsymbol{x}_0,\boldsymbol{x}_0^-,\boldsymbol{\epsilon},t}\left[-\log\left(\frac{e^{-E_\theta(\boldsymbol{x}_t,t)}}{e^{-E_\theta(\boldsymbol{x}_t,t)} + e^{-E_\theta(\boldsymbol{x}_t^-,t)}}\right)\right], \tag{2}$$

where $\boldsymbol{x}_0 \sim p(\boldsymbol{x})$, $\boldsymbol{\epsilon} \sim \mathcal{N}(\boldsymbol{0}, \boldsymbol{I})$, $\boldsymbol{x}_t = \sqrt{\bar{\alpha}_t}\boldsymbol{x}_0 + \sqrt{1-\bar{\alpha}_t}\boldsymbol{\epsilon}$. $\boldsymbol{x}_t^- = \sqrt{\bar{\alpha}_t}\boldsymbol{x}_0^- + \sqrt{1-\bar{\alpha}_t}\boldsymbol{\epsilon}$. Here $\boldsymbol{x}_0^- \sim p_{\text{corrupt}}(\boldsymbol{x}_0^-|\boldsymbol{x}_0)$ are negative examples by corrupting the positive examples $\boldsymbol{x}_0$. This contrastive loss encourages the positive (ground-truth) examples to be local energy minima.

The reverse process starts with $\boldsymbol{x}_T \sim \mathcal{N}(\boldsymbol{0}, \boldsymbol{I})$, and iteratively applies the learned denoising net $\boldsymbol{\epsilon}_\theta$, where Song et al. (2021a) introduces an adjustable noise scale $\sigma_t$:

$$\boldsymbol{x}_{t-1} = \sqrt{\bar{\alpha}_{t-1}}\frac{\boldsymbol{x}_t - \sqrt{1-\bar{\alpha}_t}\boldsymbol{\epsilon}_\theta\left(\boldsymbol{x}_t, t\right)}{\sqrt{\bar{\alpha}_t}} + \sqrt{1-\bar{\alpha}_{t-1}-\sigma_t^2}\boldsymbol{\epsilon}_\theta\left(\boldsymbol{x}_t, t\right) + \sigma_t\boldsymbol{\epsilon}_t, \ \boldsymbol{\epsilon}_t \sim \mathcal{N}(\boldsymbol{0}, \boldsymbol{I}). \tag{3}$$

Importantly, Song et al. (2021a) highlights that a diffusion model trained with $\{\alpha_t\}_{t=1}^T$ can be used to sample with $\{\alpha_{\tau_s}\}_{s=1}^S$, where $\tau$ is any increasing subsequence of $1, 2, \cdots, T$ of length $S$. This variable spacing significantly accelerates sampling and will serve as an efficient approach for MCTS simulation in Section 4.3.

## 4 METHOD

### 4.1 PROBLEM SETUP

Let $\mathcal{D} = \{\mathcal{C}, \mathcal{X}\}$ denote a dataset for a reasoning task, where each instance consists of an input $\boldsymbol{c} \in \mathbb{R}^O$ and its corresponding solution $\boldsymbol{x} \in \mathbb{R}^M$. Our goal is to model the complex reasoning problem as an optimization task to optimize an objective $\mathcal{J}(\boldsymbol{c}, \boldsymbol{x})$ and to capitalize on the diffusion model's ability to iteratively refine the solution $\boldsymbol{x}$.

In this work, we leverage an energy-based diffusion model to learn an energy function $E_\theta(\boldsymbol{c}, \boldsymbol{x}_t, t)$ to model the optimization objective $\mathcal{J}(\boldsymbol{c}, \boldsymbol{x})$. For the sake of brevity, we omit $\boldsymbol{c}$ in subsequent notations and denote it simply as $E_\theta(\boldsymbol{x}_t, t)$. During inference, we apply test-time scaling by varying the number of samples and use the learned energy function as an intrinsic verifier, thereby eliminating the need for an external verifier. However, the reasoning performance gain by test-time scaling is highly dependent on both the quality of the energy landscape and the inference-time search algorithm. To address these challenges, we propose VFScale as shown in Fig. 2, with our solutions detailed in Section 4.2 and Section 4.3.

## 4.2 TRAINING OF VFSCALE

Although the denoising loss $\mathcal{L}_{\text{MSE}}$ (Eq. 1) and the contrastive loss $\mathcal{L}_{\text{Contrast}}$ (Eq. 2) are effective, we find that simply scaling up inference budget through BoN processes yields only a marginal gain (Sec. 5.1). Through a deeper analysis, we find that the core reason is the low quality of the energy landscape, especially the lack of *performance-energy consistency*, which we address by introducing a novel Monotonic-Regression Negative Contrastive Learning (MRNCL) loss and incorporating a KL regularization, which we detail as follows.

**Monotonic-Regression Negative Contrastive Learning.** Specifically, while the contrastive loss drives positive examples to the local energy minimum, it imposes no constraints on the relative energy ordering among negative samples. For example, it is likely that a negative example $x_t^{--}$ that is further apart from the positive example $x_t$ has lower energy (higher probability) than a negative example $x_t^-$ that is nearer. Thus, the inference process can be stuck around $x_t^{--}$ and can hardly move out. This will result in reduced *performance-energy consistency*, where a lower energy at step $t$ does not necessarily correspond to a more accurate predicted solution $\hat{x}_0$, as will be shown in Sec. 5.1.

This problem cannot be easily remedied by increasing inference budget through BoN during test; instead, we explore a more fundamental way to solve this problem to shape the energy landscape by regularizing the energy among negative examples during training. The approach seeks to enforce consistency between the relative energy levels of negative samples and their corresponding performance quality, comprising the following key steps.

**(1) Generate negative samples:** Specifically, from a positive example $x_0 \sim p(x)$, we sample two negative examples $x_0^-$ and $x_0^{--}$[3], and the latter has a larger L2 distance to $x_0$. Specifically, the distances of $x_0$, $x_0^-$ and $x_0^{--}$ to $x_0$ are 0, $l_{2,0}^- = ||x_0^- - x_0||_2^2$, and $l_{2,0}^{--} = ||x_0^{--} - x_0||_2^2$, respectively.

**(2) Obtain the energy for each sample:** Then we obtain their corresponding noisy examples at step $t$ via $x_t = \sqrt{\bar{\alpha}_t}x_0 + \sqrt{1-\bar{\alpha}_t}\epsilon$, $x_t^- = \sqrt{\bar{\alpha}_t}x_0^- + \sqrt{1-\bar{\alpha}_t}\epsilon$, and $x_t^{--} = \sqrt{\bar{\alpha}_t}x_0^{--} + \sqrt{1-\bar{\alpha}_t}\epsilon$. Their energy values are $E_t^+ = E_\theta(x_t, t)$, $E_t^- = E_\theta(x_t^-, t)$, and $E_t^{--} = E_\theta(x_t^{--}, t)$, respectively.

**(3) Apply Monotonic regression to get the slope and intercept:** The core idea is to directly constrain the relationship between the energy of different samples and their distance to the ground truth using a monotonic function. Linear regression is one specific and effective instance of this approach[4], which will be used throughout the paper unless otherwise mentioned. Specifically, we fit a linear function[5] with the three points $\{(0, E_t^+), (l_{2,0}^-, E_t^-), (l_{2,0}^{--}, E_t^{--})\}$, which is characterized by the slope $k_t$ and the intercept $b_t$.

**(4) Calculate the MRNCL loss:** Then we obtain the corresponding fitted points $\{(0, \hat{E}_t^+), (l_{2,0}^-, \hat{E}_t^-), (l_{2,0}^{--}, \hat{E}_t^{--})\}$ from the linear function. We then compute the MRNCL loss as follows:

$$\mathcal{L}_{\text{MRNCL}} = \mathbb{E}_{x_0, x_0^-, x_0^{--}, \epsilon, t}\left[\max(0, \gamma - k_t) + ||E_t^+ - \hat{E}_t^+||_2^2 + ||E_t^- - \hat{E}_t^-||_2^2 + ||E_t^{--} - \hat{E}_t^{--}||_2^2\right], \quad (4)$$

where $\gamma$ is a hyperparameter and $t \sim \{0, \dots, T\}$. The first term $\max(0, \gamma - k_t)$ encourages that any three samples $x_t$, $x_t^-$, $x_t^{--}$ have a positive (and larger than $\gamma$) slope of energy vs. L2 distance. The latter three terms encourage the energy vs. L2 distance to be linear, making the energy landscape smoother.

**KL regularization.** In addition to $\mathcal{L}_{\text{MRNCL}}$ that improves *performance-energy consistency*, we further include KL-regularization (Du et al., 2021) to further improve the energy landscape:

$$\mathcal{L}_{\text{KL}} = \mathbb{E}_{t, p_{\theta,t}(x)}[E_{\text{stop-grad}(\theta)}(x)] + \mathbb{E}_{t, p_{\theta,t}(x)}[\log p_{\theta,t}(x)], \quad (5)$$

---

[3]Details to generate negative samples are in Appendix A.

[4]For its simplicity and effectiveness, we primarily use linear regression in this work and term the specific loss **LRNCL** (Linear-Regression NCL). We provide a detailed ablation study in Appendix D.2 that analyzes the impact of other monotonic constraints (e.g., quadratic, rank-based) and further justifies the choice of a linear function for achieving optimal test-time scalability.

[5]The details of linear regression algorithm can be found in Appendix A.

where $p_{\theta,t}(\boldsymbol{x})$ is the probability distribution of $\boldsymbol{x}_t$ at denoising step $t$. When optimizing w.r.t. $\mathcal{L}_{\mathrm{KL}}$, it is essentially optimizing w.r.t. the sampling (denoising) process by shaping the energy landscape to make it easier to sample. The first term in $\mathcal{L}_{\mathrm{KL}}$ encourages the samples $\boldsymbol{x}_t$ to have low energy, and the second term is maximizing the entropy of the samples, encouraging the samples to be diverse. Both terms allow better test-time scaling. Unlike Du et al. (2021), we have this KL regularization on each denoising step $t$, and we employ a more accurate estimation of entropy (Lombardi & Pant, 2016).

Together, the training objective of our VFScale is:
$$\mathcal{L} = \mathcal{L}_{\mathrm{MSE}} + \mathcal{L}_{\mathrm{Contrast}} + \mathcal{L}_{\mathrm{MRNCL}} + \mathcal{L}_{\mathrm{KL}}, \tag{6}$$
where the latter two terms improve the energy landscape and boost the test-time scalability.

## 4.3 Inference of VFScale

To fully exploit the potential of the diffusion model with a more efficient search algorithm, we propose a novel hybrid Monte Carlo Tree Search denoising (hMCTS denoising) method, which progressively applies best-of-$N$ (BoN) and Monte Carlo Tree Search (MCTS) denoising in sequence, as detailed below, throughout the denoising process. As demonstrated in Algorithm 1, we employ BoN for the diffusion process in the early diffusion stages, which introduces $L$ initial noises to maintain a consistent number of function evaluations (NFE) per example during the diffusion process.[6] Subsequently, hMCTS denoising utilizes the MCTS denoising[7] to iteratively perform the denoising process until the termination state $\boldsymbol{x}_0$ is reached. This approach enables MCTS to more accurately estimate node values when the noise is relatively small, thereby preventing the premature exclusion of potentially promising nodes. Next, we will detail the MCTS denoise process.

We treat the current diffusion state $\boldsymbol{x}_t$ as the state, the noise to remove as the action, and model the denoising process of the diffusion model as a Markov Decision Process (MDP). Therefore, a *node* in MCTS represents the state $\boldsymbol{x}_t$, along with its current visit count $N(\boldsymbol{x}_t)$ and the state value $Q(\boldsymbol{x}_t)$. A terminal node is defined as a node whose denoising step is 0. In this context, we use $\nabla_{\boldsymbol{x}_t} E_\theta(\boldsymbol{x}_t, t)$ and $E_\theta(\boldsymbol{x}_t, t)$ of the energy-based diffusion as the policy network and value network, respectively. Each deepening of the search tree corresponds to a single denoising step. Similar to MCTS in AlphaGo (Silver et al., 2016) and AlphaZero (Silver et al., 2017), each MCTS rollout consists of four core steps: selection, expansion, simulation, and backpropagation, as illustrated in Fig. 2 and Algorithm 1 in Appendix A:

**(1) Selection**: Based on the Upper Confidence Bound (UCB) from AlphaGo (Silver et al., 2016), starting from the current root node state $\boldsymbol{x}_t$, we select a child node based on the following adjusted UCB of MCTS-enhanced diffusion formula until a leaf node $\{\boldsymbol{x}_{t'}, Q(\boldsymbol{x}_{t'}), N(\boldsymbol{x}_{t'})\}$ is reached:
$$UCB(\boldsymbol{x}_t, \boldsymbol{a}_t) = Q(\boldsymbol{x}_t, \boldsymbol{a}_t) + c\sqrt{\frac{\ln N_i}{n_i}}, \tag{7}$$
where $Q(\boldsymbol{x}_t, \boldsymbol{a}_t)$ represents the value of children node, action $\boldsymbol{a}_t$ includes predicted noise $\boldsymbol{\epsilon}_\theta$ and random Gaussian noise $\boldsymbol{\epsilon}$, $n_i$ represents the number of visits to node $i$, $N_i$ represents the number of visits to the parent node of $i$, and $c$ is the exploration hyperparameter.

**(2) Expansion**: Unless the state of the node reached is a terminal state $\boldsymbol{x}_0$, we expand the children of the selected node by choosing an action and creating new nodes based on the action. For the expansion step of MCTS denoising, we perform a denoising step and add different but limited Gaussian noise. This results in $K$ distinct branches $\left\{\boldsymbol{x}_{t'-1}^{(k)} \mid k = 0, 1, \cdots, K-1\right\}$, where each child node $\boldsymbol{x}_{t'-1}^{(k)}$ is derived as the following equation adjusted from Eq. 3 :
$$\boldsymbol{x}_{t'-1}^{(k)} = \sqrt{\bar{\alpha}_{t'-1}}\frac{\boldsymbol{x}_{t'} - \sqrt{1-\bar{\alpha}_{t'}}\boldsymbol{\epsilon}_\theta\left(\boldsymbol{x}_{t'}, t'\right)}{\sqrt{\bar{\alpha}_{t'}}} + \sqrt{1 - \bar{\alpha}_{t'-1} - \sigma_{t'}^2}\boldsymbol{\epsilon}_\theta\left(\boldsymbol{x}_{t'}, t'\right) + \sigma_{t'}\boldsymbol{\epsilon}^{(k)}, \tag{8}$$
with $\boldsymbol{\epsilon}_\theta(\boldsymbol{x}_{t'}, t')$ determined by $\boldsymbol{x}_{t'}$ and $t'$, and $\boldsymbol{\epsilon}^{(k)} \sim \mathcal{N}(\boldsymbol{0}, \mathbf{I})$ representing random Gaussian noise.

---

[6]In this paper, we primarily report $K = N_r$ MCTS denoising, where ensuring $K = N_r, L = N_r$ guarantees that the NFE (Number of Function Evaluations) for each case of BoN, MCTS denoising, and hMCTS denoising remains the same. Allocating an identical NFE is the basis for a fair comparison of the search strategies' effectiveness, as detailed in Appendix F.

[7]For a detailed explanation of the distinctions between hMCTS denoising, MCTS denoising, and BoN, as well as the impact of MCTS denoising start step $t_s$, please refer to Appendix D.

**(3) Simulation**: We randomly select a child node and perform a random simulation of the MDP until a terminal state is reached. For the simulation of MCTS denoising, we use DDIM (Song et al., 2021a) for fast sampling to obtain $\hat{x}_0(x_{t'-1}^{(k^*)})$ from the randomly chosen child node state $x_{t'-1}^{(k^*)}$, and then use $E_\theta(\hat{x}_0(x_{t'-1}^{(k^*)}))$ as the reward for backpropagation instead of a reward from external verifier.[8]

**(4) Backpropagation**: Finally, we backpropagate the node values to the root node, updating the value of each node using the expected value along the path.

After $N_r$ rollouts, we select $x_{t-1}^{(k)}$ with the highest value $\frac{Q(x_{t-1}^{(k)})}{N(x_{t-1}^{(k)})}$ as the state $x_{t-1}$ for the next denoising step. The next MCTS denoising process starts from this state and proceeds to the termination state $x_0$. Here, the depth of the search tree is decided by three elements: the number of rollout steps $N_r$, the maximum number of branches $K$ for each node, and the search policy.

In summary, our proposed paradigm, VFScale, enhances *performance-energy consistency* and ease of sampling through MRNCL loss and KL regularization, respectively (Sec. 4.2), while fully unlocking the test-time scalability of diffusion models via hMCTS denoising (Sec. 4.3).

## 5 EXPERIMENTS

In the experiments, we focus on investigating the challenges of replacing an external verifier with a learned energy function to enable test-time scaling of diffusion models via varying the number of samples. We then justify the effectiveness of our innovations in both training and inference. Specifically, we aim to address the following three key questions: **(1) What factors hinder the success of test-time scaling of energy-based diffusion models? (2) Does the training of VFScale exactly improve the energy landscape to unleash the test-time scalability of diffusion models? (3) Can the inference method of VFScale achieve better test-time scaling up?**

To answer these questions, we conduct extensive experiments on two challenging reasoning tasks: Maze and Sudoku, evaluating them in **out-of-distribution (OOD) settings** where test instances are substantially more complex than the training data and cause naive methods to fail.[9] In Section 5.1, we find that the naïvely trained energy-based diffusion model (Du et al., 2024) falls short of scaling up during test time and identify the key reason as the poor quality of the energy landscape, particularly its lack of *performance-energy consistency*. Addressing this In Section 5.2, the training methods of VFScale demonstrate a significant impact on unbinding the scalability of diffusion models. As shown in Fig. 1, with the model trained with VFScale training methods, hMCTS denoising can better release the scalability, enabling generalization to much more challenging tasks during inference (Sec. 5.3).

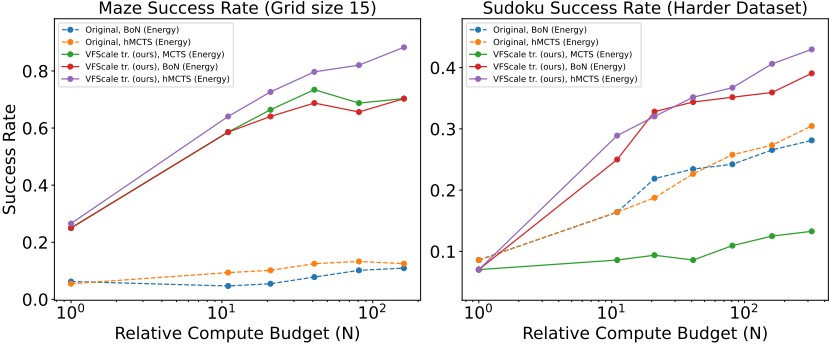

Figure 3: Scalability of different approaches on Maze and Sudoku.

---

[8]Although our method is based on an energy-based diffusion model, the proposed hMCTS is also applicable to conventional noise-predicting diffusion models, albeit requiring an external verifier. Additional results are provided in Appendix E.

[9]More experimental details can be found in Appendix B, Appendix C, and Appendix J, respectively.

## 5.1 Test-Time Scaling bottleneck of energy-based diffusion models

**Test-time scaling up the diffusion model to solve harder reasoning problems is challenging.** When tested on more challenging tasks, diffusion models trained with the original method fail to achieve substantial gains, even under larger inference budgets. Specifically, as shown in Table 1, models trained on simpler tasks with original training methods exhibit significantly poorer performance on more complex tasks. Success rate plunges from 100% on Maze size $6 \times 6$ to 6% on $15 \times 15$, and from 32% on Sudoku with 33 given entries to below 5% at 25 and 0% at 21 givens. Building upon this, we conduct best-of-N to examine whether test-time scaling can enhance the performance of the originally trained model. Table 3's "Original, BoN" row reveals that enlarging the inference budget $N$ offers marginal gains—Maze success rises by at most 5%, while Sudoku remains below 30% even at $N = 320$. These indicate that naïve test-time scaling of diffusion models fails to deliver meaningful improvements. Next, we analyze the underlying causes from the perspectives of training and inference, respectively.

Table 1: Success rate across different grid sizes of Maze and various number of given entries for naïve inference ($N = 1$) for comparison of the generalization ability of models obtained by different training methods. Let $M$ denote the grid size of Maze and $D$ denote the number of given digits in Sudoku. Original denotes the original training method in Du et al. (2024). VFScale tr. (ours) denotes our full training method, and this naming convention is used for following figures and tables. Bold font denotes the best model, and an underline denotes the second-best model. The same markings are used in the tables below.

| Methods | Maze success rate | | | | | Sudoku success rate | | | | |
|---|---|---|---|---|---|---|---|---|---|---|
| | $M=6$ | $M=8$ | $M=10$ | $M=12$ | $M=15$ | $D=33$ | $D=29$ | $D=25$ | $D=21$ | $D=17$ |
| Original | 1.0000 | 0.9062 | 0.5781 | 0.3750 | 0.0625 | 0.3203 | 0.1094 | 0.0234 | 0.0000 | 0.0000 |
| VFScale tr. w/o MRNCL | 1.0000 | 0.9922 | 0.7734 | **0.5625** | 0.2500 | **0.4219** | 0.1719 | **0.0469** | **0.0078** | 0.0000 |
| VFScale tr. w/o KL | 1.0000 | 0.9844 | 0.6953 | 0.4375 | **0.2812** | **0.4219** | **0.2578** | 0.0391 | **0.0078** | 0.0000 |
| VFScale tr. (ours) | 1.0000 | **1.0000** | **0.7750** | 0.5391 | **0.2812** | 0.1953 | 0.1016 | 0.0078 | 0.0000 | 0.0000 |

Table 2: Success rate of BoN for different training methods on Maze with grid size **15** and Sudoku harder dataset guided with ground truth. Here, $L = N$.

| Methods | Maze success rate | | | | | | Sudoku success rate | | | | | | |
|---|---|---|---|---|---|---|---|---|---|---|---|---|---|
| | $N=1$ | $N=11$ | $N=21$ | $N=41$ | $N=81$ | $N=161$ | $N=1$ | $N=11$ | $N=21$ | $N=41$ | $N=81$ | $N=161$ | $N=321$ |
| Original, BoN (Energy) | 0.0625 | 0.0469 | 0.0547 | 0.0781 | 0.1016 | 0.1094 | 0.0859 | 0.1641 | 0.2188 | 0.2344 | 0.2422 | 0.2656 | 0.2812 |
| Original, BoN (Ground Truth) | 0.0625 | 0.1250 | 0.1094 | 0.1328 | 0.1719 | 0.1719 | 0.0859 | 0.1641 | 0.2188 | 0.2344 | 0.2422 | 0.2656 | 0.2969 |
| VFScale tr. w/o MRNCL, BoN (Ground Truth) | **0.2500** | **0.5078** | **0.5938** | **0.6562** | **0.7109** | **0.7422** | **0.1094** | **0.2578** | **0.2969** | **0.3438** | **0.3750** | **0.3828** | **0.4219** |

**First, there is still considerable room for improvement on the training side to fully unlock the test-time scalability of diffusion models**[10]**.** As shown in Table 2, even when ground truth is used to guide test-time scaling, the success rate on the Maze task increases by only 7% over the energy-guided scaling method at an inference budget of $N = 161$, remaining below 20%. In the Sudoku task, raising $N$ to 321 yields only a 2% improvement compared with energy-guided scaling, still below 30%. These findings indicate that employing the learned energy function as the verifier can lead to misestimation; the *performance–energy consistency* must be further improved. Moreover, Table 4 reports that the originally trained model achieves only about 70% *performance–energy consistency* in quantitative evaluation, underscoring a significant mismatch between the energy-based verifier and actual performance. In short, these results highlight the urgent need to refine training methods to improve the quality of the energy landscape.

**Second, substantial advancements in test-time scaling methodologies are still required to fully harness the inherent scalability of diffusion models.** As shown in Table 2's first and second rows regardless of whether ground truth is as a verifier, the performance of BoN quickly plateaus. In the Maze task, increasing the inference budget $N$ from 1 to 161 raises the success rate from 6% to 17%—a gain of only 10% — and the rate of improvement slows markedly; a similar trend is observed in Sudoku, where success rate growth decelerates significantly and the maximum success

---

[10]As shown in the Appendix D.1, training an effective external verifier is non-trivial and a dense reward signal is crucial, which underscores the value of using the learned energy as an intrinsic verifier.

rate remains below 30%. These findings suggest that there remains substantial room to improve the efficiency of test-time scaling methods.

## 5.2 THE EFFECT OF VFSCALE TRAINING METHODS

Table 3: Success rate on Maze with grid size **15** and Sudoku harder dataset for comparison of the model's ability to scale up under BoN with different training methods. Here, $L = N$.

| | Maze success rate | | | | | | Sudoku success rate | | | | | | |
|---|---|---|---|---|---|---|---|---|---|---|---|---|---|
| Methods | $N$=1 | $N$=11 | $N$=21 | $N$=41 | $N$=81 | $N$=161 | $N$=1 | $N$=11 | $N$=21 | $N$=41 | $N$=81 | $N$=161 | $N$=321 |
| Original, BoN | 0.0625 | 0.0469 | 0.0547 | 0.0781 | 0.1016 | 0.1094 | 0.0859 | 0.1641 | 0.2188 | 0.2344 | 0.2422 | 0.2656 | 0.2812 |
| VFScale tr. w/o MRNCL, BoN | 0.2500 | 0.4297 | 0.5000 | 0.5391 | 0.6016 | 0.6094 | 0.1172 | 0.2656 | 0.3125 | 0.3438 | 0.3750 | 0.3906 | 0.4141 |
| VFScale tr. w/o KL, BoN | 0.2812 | 0.4688 | 0.4922 | 0.5547 | 0.5859 | 0.6250 | 0.1562 | 0.2422 | 0.2578 | 0.2656 | 0.2656 | 0.2891 | 0.3047 |
| VFScale tr. (ours), BoN | 0.2656 | 0.5859 | 0.6406 | 0.6875 | 0.6562 | 0.7031 | 0.0703 | 0.2500 | 0.3281 | 0.3438 | 0.3516 | 0.3594 | 0.3906 |

Table 4: *Performance-energy consistency* of BoN on Maze with grid size 15 to test the effect of MRNCL loss. Here, $L = N$. Details of consistency calculation can be found in Appendix A.

| | Performance-energy consistency | | | | |
|---|---|---|---|---|---|
| Methods | $N$=11 | $N$=21 | $N$=41 | $N$=81 | $N$=161 |
| Original, BoN | 0.7317 | 0.7333 | 0.7313 | 0.7283 | 0.7300 |
| VFScale tr. w/o KL, BoN | **0.8370** | **0.8445** | **0.8476** | **0.8375** | **0.8371** |

In this subsection, we evaluate the effectiveness of the MRNCL and KL losses in VFScale compared with the baseline model trained without these losses. **First,** as shown in Table 3's "N=1" columns, even without test-time scaling, integrating either the MRNCL loss or the KL loss produces significant improvements: on the Maze task, the success rate rises from 6% to 28%, and on Sudoku it likewise increases from 9% to 16%. **Second,** Table 4 also shows that adding the MRNCL loss alone produces a steady improvement in *performance–energy consistency* of over 10%. **Finally,** when we apply BoN for test-time scaling of the diffusion model, Table 3 demonstrates that using only the MRNCL loss, only the KL loss, or both, boosts the Maze success rate by more than 60% and the Sudoku success rate by over 10%. Meanwhile, as illustrated in the Fig. 4, after incorporating the MRNCL loss and the KL loss, the solutions produced by the model at different denoising steps are all noticeably closer to the ground truth solutions. Together, these findings provide strong evidence that the MRNCL and KL losses significantly enhance the test-time scalability of diffusion models.

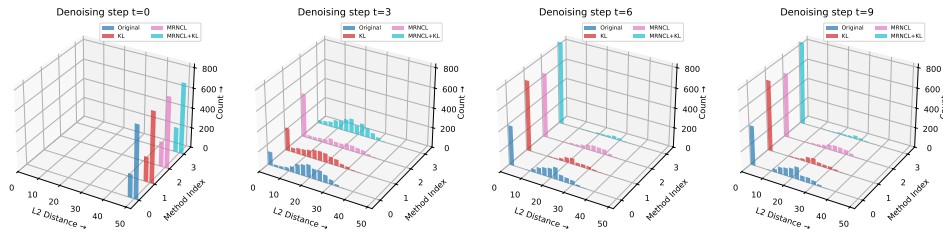

Figure 4: Comparison of the L2 distances between the solutions obtained by different training methods and the ground truth at various denoising steps.

**Trade-off between Naïve Inference and Scalability.** A critical observation from Table 1 is that the full VFScale model occasionally underperforms ablated variants (e.g., "w/o KL") in the naïve $N = 1$ setting. This is a deliberate trade-off central to our contribution. The ablated models produce "sharper" energy landscapes that may yield a confident local minimum in a single step (hence higher $N = 1$ scores) but are often riddled with poor local optima that trap search algorithms. In contrast, our joint objectives ($\mathcal{L}_{MRNCL} + \mathcal{L}_{KL}$) are explicitly designed to smooth the landscape, intentionally sacrificing $N = 1$ sharpness to ensure global navigability. The payoff for this sacrifice is superior scalability. For instance, in the Maze task (Table 3), the "w/o KL" model starts strong at 0.2812 but plateaus at 0.6250 ($N = 161$). Conversely, our full model starts lower (0.2656) but scales dramatically to 0.7031 (BoN) and further to **0.8828** with hMCTS (Table 5). Thus, the lower $N = 1$ performance is a symptom of a successful optimization for test-time search potential.

## 5.3 Test-time Scalability of VFScale

Table 5: Success rate of different approaches on Maze with grid size **15** and Sudoku harder dataset. Here, $N_r = N, K = N, L = N$.

| Methods | Maze success rate | | | | | | Sudoku success rate | | | | | | |
|---|---|---|---|---|---|---|---|---|---|---|---|---|---|
| | $N$=1 | $N$=11 | $N$=21 | $N$=41 | $N$=81 | $N$=161 | $N$=1 | $N$=11 | $N$=21 | $N$=41 | $N$=81 | $N$=161 | $N$=321 |
| Original, BoN | 0.0625 | 0.0469 | 0.0547 | 0.0781 | 0.1016 | 0.1094 | **0.0859** | 0.1641 | 0.2188 | 0.2344 | 0.2422 | 0.2656 | 0.2812 |
| Original, hMCTS denoising (ours) | 0.0625 | 0.0938 | 0.1016 | 0.1250 | 0.1328 | 0.1250 | **0.0859** | 0.1641 | 0.1875 | 0.2266 | 0.2578 | 0.2734 | 0.3047 |
| VFScale tr. (ours), MCTS denoising (ours) | 0.2656 | 0.5859 | 0.6641 | 0.7344 | 0.6875 | 0.7031 | 0.0703 | 0.0859 | 0.0938 | 0.0859 | 0.1094 | 0.1250 | 0.1328 |
| VFScale tr. (ours), BoN | 0.2656 | 0.5859 | 0.6406 | 0.6875 | 0.6562 | 0.7031 | 0.0703 | 0.2500 | 0.3281 | 0.3438 | 0.3516 | 0.3594 | 0.3906 |
| VFScale tr. (ours), hMCTS denoising (ours) | **0.2656** | **0.6406** | **0.7266** | **0.7969** | **0.8203** | **0.8828** | 0.0703 | **0.2891** | 0.3203 | **0.3516** | **0.3672** | **0.4062** | **0.4297** |

Using the diffusion model trained with VFScale training methods, we evaluate various inference approaches to validate the efficacy of our inference methods described in Section 4.3. As shown in Table 5, in the Maze experiment, the MCTS denoising method slightly outperforms BoN, while our hMCTS denoising yields a significantly higher success rate, with a maximum improvement of approximately 18% than BoN. Moreover, as budget $N$ increases, the performance gap between hMCTS denoising and BoN widens. In the Sudoku experiment, hMCTS denoising also consistently outperforms BoN, with a maximum improvement of 5%. As illustrated in the *scaling curve* in Fig. 3, hMCTS denoising with the model trained with additional MRNCL loss and KL loss shows a marked improvement compared to both other inference methods. At the same time, the rate of improvement for other inference methods clearly slows down compared to hMCTS denoising. These results provide strong evidence that our inference method can effectively scale up during test time, offering a clear advantage over BoN.

## 6 Conclusion

In this work, we have introduced the Verifier-free Test-time Scalable Diffusion Model (VFScale), a novel framework to achieve scalable intrinsic reasoning to tackle more complex reasoning tasks than in training. VFScale explores applying the varying number-of-sample scaling method with learned energy as a verifier. Concretely, faced with a low-quality energy landscape and the lack of efficient search algorithms, VFScale is composed of two innovations in training and inference to address these challenges. On the training side, VFScale introduces two auxiliary training losses, $\mathcal{L}_{\text{MRNCL}}$ and $\mathcal{L}_{\text{KL}}$, to improve the energy landscape. This, in turn, ensures that the learned energy function is an effective verifier for test-time scaling. On the inference side, VFScale integrates hybrid Monte Carlo Tree Search (hMCTS) denoising to better leverage the model's test-time scalability. We have conducted extensive experiments on Maze and Sudoku to validate the efficacy of VFScale and believe that VFScale offers a robust solution for test-time scaling, unlocking the notable potential of diffusion models for complex reasoning. Limitations and future directions are discussed in Appendix I.

## Acknowledgments

This work was supported by the Westlake University Center for High-performance Computing. The views and conclusions contained herein are those of the authors and should not be interpreted as representing the official policies or endorsements of the funding entities.

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

# Part I

# Appendix

## Table of Contents

# A    RELATED ALGORITHMS AND METRIC CALCULATION

## A.1    PERFORMANCE-ENERGY CONSISTENCY

From a high-level perspective, higher performance–energy consistency indicates that the learned energy function serves as a more accurate intrinsic verifier; conversely, lower consistency denotes less precise verification. In this paper, performance-energy consistency refers to the consistency between the results evaluated using an energy model and those evaluated using real-world metrics for the same sample. Specifically, the consistency requires that good samples are assigned low energy, while poor samples are assigned high energy. Performance-energy consistency measures the proportion of element pairs that maintain the same relative order in both permutations $X$ and $Y$, where $X$ and $Y$ represent the index arrays obtained by sorting the original energy values $\mathbf{E} = (E_1, E_2, \ldots, E_N)$ and performance metric values $\mathbf{P} = (P_1, P_2, \ldots, P_N)$, respectively, in ascending order. In this paper, the energy values are calculated by energy model $E_\theta(x_0)$ for samples $x_0$. The performance metric values are calculated as the L2 distance between the generated samples $x_0$ and the ground truth under the given condition.

Let $X = (X_1, X_2, \ldots, X_N)$ and $Y = (Y_1, Y_2, \ldots, Y_N)$ be the index arrays obtained by sorting the original energy values $\mathbf{E} = (E_1, E_2, \ldots, E_N)$ and performance metric values $\mathbf{P} = (P_1, P_2, \ldots, P_N)$, respectively, in ascending order. Specifically, $X_i$ is the rank of the $i$-th sample in the sorted energy values $\mathbf{E}$, and $Y_i$ is the rank of the $i$-th sample in the sorted performance metric values $\mathbf{P}$.

**Consistency Definition:** The **consistency** is defined as the proportion of consistent pairs $(i, j)$ where $i < j$ and the relative order of $i$ and $j$ in $X$ is the same as in $Y$. Specifically:

$$\text{Consistency} = \frac{1}{\binom{N}{2}} \sum_{i=1}^{N-1} \sum_{j=i+1}^{N} \mathbb{I}\left((X_i < X_j \wedge Y_i < Y_j) \vee (X_i > X_j \wedge Y_i > Y_j)\right),$$

where:

- $\binom{N}{2} = \frac{N(N-1)}{2}$ is the total number of pairs $(i, j)$ with $i < j$,

- $\mathbb{I}[\cdot]$ is the indicator function, which evaluates to 1 if the condition inside the brackets holds (i.e., the relative order is consistent), and 0 otherwise.

## A.2    NEGATIVE SAMPLE GENERATION

Negative samples are generated by introducing noise into the positive sample $x_0$. In the Maze and Sudoku experiments, permutation noise is applied to the channel dimension to induce significant changes in the solution. Other noise types can be used, as this remains a hyperparameter choice. Specifically, we first randomly sample two scalars $p_1$ and $p_2$ from a uniform distribution in the interval $[0, 1]$, i.e., $p_1, p_2 \sim \text{Uniform}(0, 1)$ ($p_1 < p_2$). Then, for each channel position of the positive sample $x_0$, we swap the channel positions with probabilities $p_1$ and $p_2$, resulting in $x_0^-$ and $x_0^{--}$, such that the L2 distance between $x_0^-$ and $x_0$ is smaller than the L2 distance between $x_0^{--}$ and $x_0$. For other noise types, such as Gaussian noise, we normalize the L2 norm of the noise and apply noise at different scales to ensure that the L2 distance from $x_0^-$ to $x_0$ is smaller than the L2 distance from $x_0^{--}$ to $x_0$.

## A.3    LINEAR-REGRESSION ALGORITHM

Given three points $(x_1, y_1)$, $(x_2, y_2)$, and $(x_3, y_3)$, we wish to fit a line of the form Lane et al. (2003):

$$y = kx + b$$

The mean of the $x$-coordinates and the mean of the $y$-coordinates are:

$$\bar{x} = \frac{1}{3}(x_1 + x_2 + x_3), \quad \bar{y} = \frac{1}{3}(y_1 + y_2 + y_3)$$

The slope $k$ of the best-fit line is given by the formula:

---

**Algorithm 1** hMCTS denoising of VFScale

---

1: **Input:** EBM $E_\theta(\cdot)$, Diffusion Steps $T$, MCTS denoising start step $t_s$, Number of initial noise for Best-of-N $L$, Maximum MCTS branch count $K$, Maximum MCTS rollout step $N_r$, A set of initial diffusion state $\{x_T^{(k)} \mid k = 1, ..., L\}$ sampled from Gaussian noise.
2: // BoN for $T \to t_s$
3: **for** $t = T$ **to** $t_s$ **do**
4:  Use Eq. 3 to get $x_{t-1}^{(k)}$ for $k = 1, ..., L$
5: **end for**
  $x_{t_s} \leftarrow \arg\min_{x_{t_s}^{(k)}} E_\theta(x_{t_s}^{(k)})$
6: // MCTS denoising for $t_s \to 1$
7: **for** $t = t_s$ **to** 1 **do**
8:   // Do MCTS Rollouts:
9:   **for** $i = 1$ **to** $N_r$ **do**
10:    **Selection:** Use UCB from Eq. 7 to select the child node until a leaf node or a terminal node $\{x_{t'}, Q(x_{t'}), N(x_{t'})\}$ is reached and form a path using the nodes accessed during the selection;
11:    **Expansion:** Use Eq. 8 to generate child node state $x_{t'-1}^{(k)}$ for $x_{t'}$ and initialize $Q(x_{t'-1}^{(k)}) = 0$, $N(x_{t'-1}^{(k)}) = 0$ for $k = 0, ..., K-1$;
12:    **Simulation:** Randomly choose $k^* \sim \{0, ..., K-1\}$ and do DDIM simulation from $x_{t'-1}^{(k^*)}$ to get $\hat{x}_0(x_{t'-1}^{(k^*)})$;
13:    **Backpropagation:** Update the value and visit count of each node $x_{t_p}$ in the path using $Q(x_{t_p}) \leftarrow Q(x_{t_p}) - E_\theta(\hat{x}_0(x_{t'-1}^{(k^*)}))$;
14:      $N(x_{t_p}) \leftarrow N(x_{t_p}) + 1$;
15:   **end for**
16:   $x_{t-1} \leftarrow \arg\max_{x_{t-1}^{(k)}} \frac{Q(x_{t-1}^{(k)})}{N(x_{t-1}^{(k)})}$
17: **end for**
18: **return** $x_0$

---

$$k = \frac{\sum_{i=1}^{3}(x_i - \bar{x})(y_i - \bar{y})}{\sum_{i=1}^{3}(x_i - \bar{x})^2}$$

This formula represents the least-squares solution for the slope. Once the slope $k$ is determined, the intercept $b$ can be calculated as:

$$b = \bar{y} - k\bar{x}$$

The equation of the best-fit line is:

$$\hat{y} = kx + b$$

### A.4   INFERENCE ALGORITHM

The detailed algorithm of hMCTS of VFScale inference method is in Algorithm 1.

## B   DETAILS OF EXPERIMENTS

### B.1   OVERVIEW

In Maze experiments, the datasets are generated by Ivanitskiy et al. (2023), and the diffusion model is mainly trained with Maze sizes of up to $6 \times 6$ while tested on harder datasets with sizes significantly larger than $6 \times 6$ (Fig. 1). For Sudoku experiments, the basic setting is adopted from Du et al. (2024) where the diffusion model is trained on SAT-Net dataset with 31 to 42 given entries Wang et al. (2019) and tested on the harder RRN dataset with 17 to 34 given entries Palm et al. (2018) with fewer given entries. All models and inference methods are evaluated with solving success rate. Here successful

solving means the predicted solution exactly matches the ground-truth solution on all entries, a very stringent metric[11].

All training methods are compared with the original training pipeline in Du et al. (2024). LRNCL and KL represent the loss terms $\mathcal{L}_{\text{LRNCL}}$ and $\mathcal{L}_{\text{KL}}$, respectively in Eq. 6. VFScale tr. w/o LRNCL, VFScale tr. w/o KL, and VFScale tr. (ours) represent the three training methods of VFScale, the last being the full version. **The MCTS denoising and hMCTS denoising are compared with BoN with the same computational budget.** The experiment code can be found at the repo.

## B.2 CORE METRIC DEFINITION

For Maze, successful solving means finding a continuous path from starting location to target location without breaking or overlapping with the wall; for Sudoku, successful solving means filling all the missing numbers that *exactly* match the ground truth, both of which are very stringent metrics.

## B.3 DETAILS OF SUDOKU EXPERIMENTS

For Sudoku experiment, the dataset, model architecture, and training configurations are adopted from Du et al. (2024). We mainly use solving success rate to evaluate different models. The model backbone and training configurations can be found in Fig. 5 and Table 6, respectively. All exploration hyperparameters $c$ are set as 100 for the Sudoku task.

| |
|---|
| 3x3 Conv2D, 384 |
| Resblock 384 |
| Resblock 384 |
| Resblock 384 |
| Resblock 384 |
| Resblock 384 |
| Resblock 384 |
| 3x3 Conv2D, 9 |

Figure 5: The model architecture for VFScale on Sudoku task. The energy value is computed using the L2 norm of the final predicted output similar to Du et al. (2023), while the output is directly used as noise prediction for the diffusion baseline.

Table 6: **Details of training for Sudoku task**.

| Training configurations | |
|---|---|
| Number of training steps | 100000 |
| Training batch size | 64 |
| Learning rate | 0.0001 |
| Diffusion steps | 10 |
| Inner loop optimization steps | 20 |
| Denoising loss type | MSE |
| Optimizer | Adam |

## B.4 DETAILS OF MAZE EXPERIMENTS

The details of maze experiments and the model backbone are provided in Table 7 and Fig. 6, respectively. The key metric, the maze-solving success rate, is defined as the proportion of model-generated paths that have no breakpoints, do not overlap with walls, and begin and end at the start

---

[11]For example, a successful solving of a $15 \times 15$ Maze needs to predict all $31 \times 31 = 961$ entries correctly (for each grid cell, predict path/not path/wall), where the path length from start to target is on the order of $10^2$.

and target points, respectively. Maze datasets are generated by Ivanitskiy et al. (2023), and detailed hyperparameter configurations are in Table 7. All the exploration hyperparameters $c$ are set as 100 for Maze task.

| |
| :---: |
| 3x3 Conv2D, 384 |
| Resblock 384 |
| Resblock 384 |
| Resblock 384 |
| Resblock 384 |
| Resblock 384 |
| Resblock 384 |
| 3x3 Conv2D, 9 |

Figure 6: The model architecture for VFScale on Maze task. The energy value is computed using the L2 norm of the final predicted output similar to Du et al. (2023), while the output is directly used as noise prediction for the diffusion baseline.

Table 7: **Details of Maze dataset, training**.

| Dataset: | |
| :--- | :---: |
| Size of training dataset with grid size 4 | 10219 |
| Size of training dataset with grid size 5 | 9394 |
| Size of training dataset with grid size 6 | 10295 |
| Minimum length of solution path | 5 |
| Algorithm to generate the maze | DFS |
| Size of test dataset with grid size 6 | 837 |
| Size of test dataset with grid size 8 | 888 |
| Size of test dataset with grid size 10 | 948 |
| Size of test dataset with grid size 12 | 960 |
| Size of test dataset with grid size 15 | 975 |
| Size of test dataset with grid size 20 | 978 |
| Size of test dataset with grid size 30 | 994 |
| Training configurations | |
| Number of training steps | 200000 |
| Training batch size | 64 |
| Learning rate | 0.0001 |
| Diffusion steps | 10 |
| Inner loop optimization steps | 20 |
| Denoising loss type | MSE + MAE |
| Optimizer | Adam |

## B.5 COMPUTE RESOURCES

For training efficiency, we report the training time of our models along with the machine information (1 GPU with 80 GB of VRAM, CPU: 16). On the Maze (100,000 steps) environment, naive training took 2 hours, while incorporating LRNCL increased the training time to 4 hours. Adding KL regularization resulted in a 3-hour training time, and combining LRNCL and KL required 6 hours. For the Sudoku (200,000 steps) environment, naive training took 4 hours. Training with +LRNCL took 5 hours, +KL took 6 hours, and +LRNCL+KL required 7 hours.

## B.6 DATA REPRESENTATION AND PROCESSING FOR DISCRETE TASKS

A key aspect of our "continuous reasoning formulation" is its application to discrete tasks like Maze (and Sudoku) via a discrete-continuous-discrete pipeline. This approach, which is common

for applying generative models to discrete data (Du et al., 2024), allows the model to leverage gradient-based refinement in a continuous space. The process is as follows:

- **Discrete → Continuous Encoding:** The discrete $H \times W$ Maze task is first encoded into a continuous tensor. We treat each grid cell as having one of $C$ states (e.g., "path," "wall," "empty") and use a one-hot encoding for each cell. This transforms the discrete $H \times W$ grid into a continuous $H \times W \times C$ tensor, $x_0$, which serves as the model's ground truth.

- **Continuous Diffusion Process:** Our energy function, $E_\theta$, and all search algorithms (BoN/hMCTS) operate entirely in this continuous $H \times W \times C$ space. All diffusion, noise prediction, and energy calculations are performed on these continuous-valued tensors.

- **Visualization (e.g., in Figure 2):** To clarify our overview figure, the intermediate states shown (e.g., $x_k$) are not discrete paths. They are renderings of the continuous probability distribution (e.g., in the "path" channel) of the predicted $\hat{x}_0$ at an intermediate step. This visualizes the model's evolving "belief" as it denoises from $t = T$ (pure noise) to $t = 0$ (a confident solution).

- **Decoding to Discrete:** To obtain the final discrete solution for evaluation, we apply a standard `argmax` operation along the channel dimension ($C$) of the final output tensor $\hat{x}_0$. This "collapses" the probability distribution back to the single most likely discrete state for each grid cell. This final discrete grid is then evaluated for success.

## C    PERFORMANCE SENSITIVITY TO HYPERPARAMETERS

In this subsection, we analyze the impact of several hyperparameters on the experimental results. As shown in Table 9, the influence of different noise scales on the performance of various methods is presented. The hMCTS denoising and BoN require a relatively larger noise scale to better expand the search space and improve final performance, while the diffusion model with naive inference performs best with a smaller noise scale. As demonstrated in Table 8 and Fig. 7, the effect of varying inner-loop optimization steps on the results is also analyzed. It can be observed that performance improves gradually with an increasing number of steps, and after 5 steps, the performance stabilizes and the improvement slows down. Therefore, we chose 5 inner-loop optimization steps for the Maze experiments.

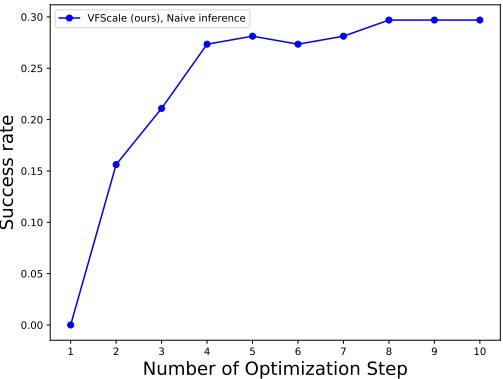

Figure 7: Visualization of success rate across different number of inner-loop optimization steps on Maze with grid size $15 \times 15$.

Table 8: Success rate across the different number of inner-loop optimization step on Maze with grid size **15**.

| Methods | Number of optimization step | | | | | | | | | |
|---|---|---|---|---|---|---|---|---|---|---|
| | 1 | 2 | 3 | 4 | 5 | 6 | 7 | 8 | 9 | 10 |
| VFScale tr. (ours), Naive inference | 0.0000 | 0.1562 | 0.2109 | 0.2734 | 0.2812 | 0.2734 | 0.2812 | 0.2969 | 0.2969 | 0.2969 |

Table 9: Success rate across different noise scales on Maze with grid size **15**.

| Methods | Noise scale | | | | | | | | | |
|---|---|---|---|---|---|---|---|---|---|---|
| | 0.1 | 0.2 | 0.3 | 0.4 | 0.5 | 0.6 | 0.7 | 0.8 | 0.9 | 1.0 |
| VFScale tr. (ours), hMCTS denoising (energy) | 0.3828 | 0.4375 | 0.5312 | 0.6094 | 0.6562 | 0.6953 | 0.7031 | 0.7344 | 0.7734 | 0.7969 |
| VFScale tr. (ours), naive inference | 0.3125 | 0.2656 | 0.2578 | 0.2344 | 0.2422 | 0.2656 | 0.2578 | 0.2422 | 0.2500 | 0.2500 |
| VFScale tr. (ours), BoN(energy) | 0.3906 | 0.4453 | 0.5312 | 0.5703 | 0.5938 | 0.6328 | 0.6641 | 0.6719 | 0.6797 | 0.6562 |

# D  ADDITIONAL RESULTS

## D.1  THE CHALLENGE OF EXTERNAL VERIFIERS AND THE VALUE OF INTRINSIC ENERGY

In the main text, we argue that leveraging the learned energy function as an **intrinsic verifier** is a cornerstone of our verifier-free, test-time scaling framework. This subsection provides a detailed comparative analysis to substantiate this claim, demonstrating two key points: (1) training a high-performing **external verifier** for complex reasoning tasks is nontrivial, and (2) the **dense, continuous reward signal** provided by our intrinsic energy is crucial for effective scaling and performs on par with a perfect oracle.

**Intrinsic Verifier vs. Learned External Verifier:** To assess the difficulty and effectiveness of using a separately trained model as a verifier, we constructed a **learnable external verifier**. Specifically, we trained a classifier to predict the element-wise accuracy of a given Maze solution. While this classifier achieved a high correlation of 0.99 with the true ground truth scores, its performance as a guide for test-time scaling was substantially inferior to our intrinsic energy verifier.

As shown in Table 10, when using Best-of-N (BoN) sampling, the external verifier leads to a performance drop of up to 30% compared to using the intrinsic energy function. We attribute this significant gap to the external verifier's inability to distinguish between the subtle yet critical differences among high-quality candidate solutions generated during the scaling process, leading to suboptimal selections. This experiment highlights that even a verifier with high statistical correlation may fail to provide the precise guidance needed for complex reasoning.

Table 10: Comparison of test-time scaling performance on the $15 \times 15$ Maze task using our **intrinsic energy verifier** versus a trained **external verifier**. Both methods use the same set of generated samples under the Best-of-N (BoN) framework. Success rates are reported for different compute budgets ($N$).

| Method | N=1 | N=11 | N=21 | N=41 | N=81 |
|---|---|---|---|---|---|
| VFScale tr. (ours), BoN, Intrinsic Verifier | 0.2656 | 0.5859 | 0.6406 | 0.6875 | 0.6562 |
| VFScale tr. (ours), BoN, External Verifier | 0.2656 | 0.3516 | 0.3594 | 0.3438 | 0.3203 |

**The Importance of a Dense Reward Signal:** To further locate the advantage of our intrinsic verifier, we compared its performance against two perfect ground-truth (GT) oracles:

- A **sparse GT 0/1 verifier**, which provides a binary success/failure signal only.
- A **continuous GT score verifier**, which provides a precise continuous quality score for any given sample (a perfect, dense reward).

The results in Table 11 are highly informative. First, the sparse 0/1 verifier performs significantly worse (over a 10% gap at $N = 81$) than our dense, energy-guided method. This starkly illustrates the

advantage of continuous guidance; a dense reward signal allows the search algorithm (hMCTS) to make more informed decisions at each step, rather than relying on a simple binary outcome.

Crucially, the performance of our intrinsic verifier is **nearly identical** to that of the perfect continuous GT score verifier. This result strongly validates the effectiveness of our training objectives (especially LRNCL) in shaping a high-quality energy landscape that accurately reflects the true solution quality. In essence, our framework successfully learns an intrinsic verifier that functions as a near-perfect, dense reward oracle, unlocking the full potential of test-time scaling without external supervision.

Table 11: Performance comparison on the $15 \times 15$ Maze task between our energy-guided **intrinsic verifier** and two perfect ground-truth (GT) oracles: a **sparse binary verifier** and a **continuous score verifier**. All methods use the hMCTS search algorithm.

| Method | N=1 | N=11 | N=21 | N=41 | N=81 |
|---|---|---|---|---|---|
| VFScale (ours), hMCTS, **Energy Guided (Intrinsic)** | 0.2656 | 0.6406 | 0.7266 | 0.7969 | 0.8203 |
| VFScale (ours), hMCTS, **GT 0/1 Guided (Sparse)** | 0.2656 | 0.5469 | 0.5938 | 0.6562 | 0.6719 |
| VFScale (ours), hMCTS, **GT Score Guided (Dense)** | 0.2656 | 0.6328 | 0.7266 | 0.7734 | 0.8359 |

## D.2 ABLATION STUDY ON MONOTONIC REGRESSION CONSTRAINTS FOR MRNCL

To validate the effectiveness of the linear constraint and analyze the impact of different energy-L2 distance relationships within our Monotonic Regression Negative Contrastive Learning (MRNCL) framework, we conducted a comprehensive ablation study on the Maze task. We compare our proposed **LRNCL** (Linear) against several variants:

- **QRNCL (Quadratic Regression NCL):** This method fits a quadratic function, encouraging the energy-L2 distance of samples to lie within its monotonically increasing region.

- **RBNCL (Rank-based NCL):** This uses the rank correlation between sample energies and their L2 distances as a loss, without assuming a specific functional form for the relationship.

- **FTNCL (Fixed-transformation NCL):** This requires the learned energy function to approximate a pre-defined linear function with fixed, non-adaptive parameters.

Table 12: Success rates on the $15 \times 15$ Maze task for different MRNCL variants. While QRNCL achieves the best performance for single-shot generation (N=1), **LRNCL demonstrates superior test-time scalability** as the compute budget ($N$) increases, achieving the highest success rate at N=81.

| Training Method | N=1 | N=11 | N=21 | N=41 | N=81 |
|---|---|---|---|---|---|
| Original, BoN, Energy Guided | 0.0625 | 0.0469 | 0.0547 | 0.0781 | 0.1016 |
| **+LRNCL (ours)** | 0.2812 | **0.4688** | **0.4922** | **0.5547** | **0.5859** |
| +QRNCL | **0.3516** | 0.3672 | 0.3906 | 0.3984 | 0.3828 |
| +RBNCL | 0.1875 | 0.4219 | 0.4766 | 0.5234 | 0.5781 |
| +FTNCL | 0.3125 | 0.4141 | 0.4844 | 0.4922 | 0.5156 |

The scaling results in Table 12 show that all MRNCL variants significantly improve performance over the baseline. Notably, an interesting trade-off emerges: QRNCL yields the largest performance gain without scaling (N=1), but offers the weakest benefit for test-time scalability, with the performance even degrading at N=81. In contrast, RBNCL exhibits strong scalability, surpassed only by LRNCL, which provides the largest and most consistent scalability gain.

These findings collectively demonstrate that our proposed LRNCL, chosen for its simplicity, is highly effective at calibrating the energy landscape to improve a model's test-time scalability, which is the primary goal of this work.

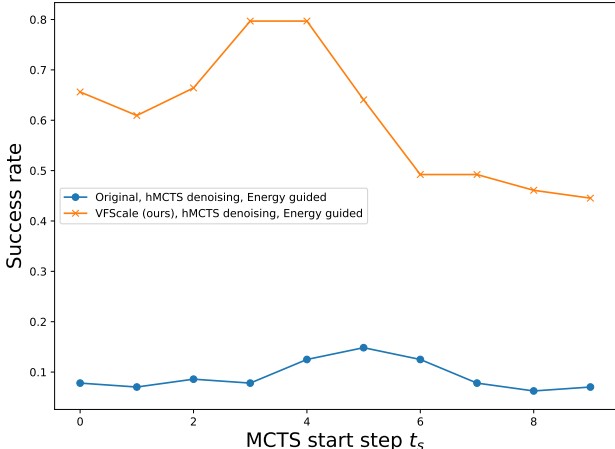

Figure 8: Visualization of Success rate across different MCTS start step $t_s$.

### D.3 ANALYSIS ON SWITCH-OVER STEP $t_s$ OF HMCTS

The parameter $t_s$ controls the proportion of the total inference budget allocated to MCTS denoising. When $t_s = 9$, it means only MCTS denoising is used, while $t_s = 0$ means only BoN is used. For $0 < t_s < 9$ (**maximum denosing step is 10.**), hMCTS denoising is applied. As shown in Table 14 and Fig. 8, there is a noticeable peak in model performance as $t_s$ varies.

While a manually tuned $t_s$ is effective, we also developed an **adaptive mechanism** to set the switch-over step on a per-sample basis, eliminating the need for hyperparameter tuning. This method achieves a more dynamic exploration-exploitation balance by using Best-of-N (BoN) for early-stage global exploration and switching to MCTS for later-stage local exploitation based on the state of the search.

Specifically, the mechanism utilizes the energy variance across the different search branches. The switch from BoN to MCTS occurs when this energy variance exceeds a preset threshold, creating a closed-loop system that dynamically balances the two strategies. As shown in Table 13, this adaptive approach shows a slight advantage over a manually tuned $t_s$, particularly with smaller inference budgets (e.g., at N=11 and N=21), demonstrating its utility and robustness.

Table 13: Success rate on the $15 \times 15$ Maze task comparing a manually tuned, fixed switch-over step ($t_s$) with our proposed **adaptive** $t_s$ mechanism. The adaptive method shows a notable advantage at smaller compute budgets.

| Switch-Over Method | N=11 | N=21 | N=41 | N=81 |
|---|---|---|---|---|
| VFScale tr. (ours), hMCTS, Tuned $t_s$ | 0.6406 | 0.7266 | 0.7969 | 0.8203 |
| VFScale tr. (ours), hMCTS, Adaptive $t_s$ | 0.7031 | 0.7734 | 0.7891 | 0.8203 |

Table 14: Success rate of hMCTS denoising on Maze with grid size **15** across different MCTS start steps.

| Methods | 0 | 1 | 2 | 3 | 4 | 5 | 6 | 7 | 8 | 9 |
|---|---|---|---|---|---|---|---|---|---|---|
| Original, hMCTS denoising (energy) | 0.0781 | 0.0703 | 0.0859 | 0.0781 | 0.1250 | 0.1484 | 0.1250 | 0.0781 | 0.0625 | 0.0703 |
| VFScale tr. (ours), hMCTS denoising (energy) | 0.6562 | 0.6094 | 0.6641 | 0.7969 | 0.7969 | 0.6406 | 0.4922 | 0.4922 | 0.4609 | 0.4453 |

Table 15: Success rate of BoN for different training methods on Maze with grid size **15** and Sudoku harder dataset guided with ground truth accuracy. Untrained, BoN (gt) represents using ground truth to guide the BoN. Here, $L = N$. Bold font denotes the best model.

| | Maze success rate | | | | | | Sudoku success rate | | | | | | |
|---|---|---|---|---|---|---|---|---|---|---|---|---|---|
| Methods | $N$=1 | $N$=11 | $N$=21 | $N$=41 | $N$=81 | $N$=161 | $N$=1 | $N$=11 | $N$=21 | $N$=41 | $N$=81 | $N$=161 | $N$=321 |
| Untrained, BoN (gt) | 0.0000 | 0.0000 | 0.0000 | 0.0000 | 0.0000 | 0.0000 | 0.0000 | 0.0000 | 0.0000 | 0.0000 | 0.0000 | 0.0000 | 0.0000 |
| Original, BoN (gt) | 0.0625 | 0.1250 | 0.1094 | 0.1328 | 0.1719 | 0.1719 | 0.0859 | 0.1641 | 0.2188 | 0.2344 | 0.2422 | 0.2656 | 0.2969 |
| DDPM, BoN (gt) | 0.0312 | 0.1094 | 0.1587 | 0.1746 | 0.2031 | 0.2422 | 0.0000 | 0.0000 | 0.0000 | 0.0000 | 0.0000 | 0.0000 | 0.0156 |
| VFScale tr. w/o LRNCL, BoN (gt) | **0.2500** | **0.5078** | **0.5938** | **0.6562** | **0.7109** | **0.7422** | **0.1094** | **0.2578** | **0.2969** | **0.3438** | **0.3750** | **0.3828** | **0.4219** |

Table 16: Success rate and element-wise accuracy of BoN for different training methods on Sudoku harder dataset guided with ground truth accuracy. Here, $L = N$. Bold font denotes the best model.

| | Success rate | | | | | | | Element-wise accuracy | | | | | | |
|---|---|---|---|---|---|---|---|---|---|---|---|---|---|---|
| Methods | $N$=1 | $N$=11 | $N$=21 | $N$=41 | $N$=81 | $N$=161 | $N$=321 | $N$=1 | $N$=11 | $N$=21 | $N$=41 | $N$=81 | $N$=161 | $N$=321 |
| DDPM, BoN, GT accuracy guided | 0.0000 | 0.0000 | 0.0000 | 0.0000 | 0.0000 | 0.0000 | 0.0156 | 0.5071 | 0.6089 | 0.6316 | 0.6492 | 0.6691 | 0.6881 | 0.6999 |
| Original, BoN, GT accuracy guided | 0.0781 | 0.1641 | 0.2188 | 0.2344 | 0.2422 | 0.2656 | 0.2812 | **0.6650** | 0.7731 | 0.7952 | 0.8036 | 0.8217 | 0.8347 | 0.8491 |
| VFScale tr. w/o LRNCL, BoN, GT accuracy guided | **0.1094** | **0.2578** | **0.2969** | **0.3438** | **0.3750** | **0.3828** | **0.4219** | 0.6442 | **0.7855** | **0.8096** | **0.8317** | **0.8466** | **0.8628** | **0.8854** |

## D.4 Impact of Negative Sample Generation Strategy

To analyze the sensitivity of our framework to the negative sampling strategy, we compare the permutation-based approach used in our main experiments against a more standard Gaussian noise perturbation method. In this alternative, negative samples are generated by adding scaled Gaussian noise to the positive samples.

The results presented in Table 17 lead to two key findings. First, the LRNCL framework robustly improves performance and unlocks test-time scalability regardless of the specific negative sample generation strategy employed, demonstrating its general applicability. Both perturbation methods significantly outperform the original baseline.

Second, the choice of strategy is not without impact. While the Gaussian noise approach provides a strong performance boost, the permutation-based method yields superior test-time scalability, with its advantage growing as the compute budget ($N$) increases. This suggests that structured, problem-aware perturbations (like permutation noise for grid-based tasks) can create more informative negative samples for training the energy landscape compared to unstructured noise.

Table 17: Success rate on the $15 \times 15$ Maze task comparing different negative sample generation strategies for LRNCL. While both strategies significantly outperform the baseline, the **Permutation-based** method shows better scalability.

| Training Method | N=1 | N=11 | N=21 | N=41 | N=81 |
|---|---|---|---|---|---|
| Original, BoN, Energy Guided | 0.0625 | 0.0469 | 0.0547 | 0.0781 | 0.1016 |
| Permutation-based Perturb (ours) | 0.2812 | **0.4688** | **0.4922** | **0.5547** | **0.5859** |
| Gaussian Noise Perturb | **0.3125** | 0.3750 | 0.3906 | 0.4062 | 0.4141 |

## D.5 Stability Analysis and Multi-Seed Results

To validate the stability of our framework, we computed the **standard deviation over 10 runs with different random seeds** for our key experiments on the $15 \times 15$ Maze task. The results, presented in Table 18, summarize the performance (mean $\pm$ std) for the primary methods as the compute budget (N) increases.

The data confirms that our VFScale framework is not only superior in performance but also **stable**, as indicated by the tight standard deviations. Both our training (VFScale tr.) and inference (hMCTS) methods show significant and consistent gains over the baseline, reinforcing the robustness of our approach.

Table 18: Mean and standard deviation of success rates on the $15 \times 15$ Maze task over 10 random seeds.

| Methods | N=11 | N=21 | N=41 | N=81 |
|---|---|---|---|---|
| Original, BoN | $0.0539 \pm 0.0055$ | $0.0594 \pm 0.0087$ | $0.0594 \pm 0.0063$ | $0.0609 \pm 0.0058$ |
| VFScale tr. (ours), BoN | $0.6203 \pm 0.0230$ | $0.6562 \pm 0.0148$ | $0.6680 \pm 0.0176$ | $0.6836 \pm 0.0165$ |
| **VFScale tr. (ours), hMCTS (ours)** | **$0.6727 \pm 0.0230$** | **$0.7336 \pm 0.0180$** | **$0.7945 \pm 0.0086$** | **$0.8422 \pm 0.0147$** |

## D.6 FROM LOCAL CONSTRAINTS TO GLOBAL RANKING ALIGNMENT

A natural question arises regarding our training objective: since MRNCL enforces only *local* monotonicity (between a positive sample and its perturbed negatives), how does this translate to a *globally* aligned energy landscape? Our hypothesis is that by repeatedly enforcing this simple, local, linear consistency across billions of samples, batches, and diffusion timesteps during training, we implicitly shape a globally structured and reliable energy landscape. We provide both direct and indirect evidence to support this.

**Direct Evidence: Global Consistency Metrics.** First, our Performance-Energy Consistency (PEC) metric (defined in Appendix A) serves as a measure of global rank-order consistency across a batch of samples. As shown in Table 4, our training improves this global consistency metric from $\sim 73\%$ (Original) to $\sim 84\%$ (VFScale).

To further rigorously quantify this, we conducted a new experiment measuring the **Kendall-$\tau$ rank correlation**—a standard metric for global ranking alignment—between the energy values and the true solution quality across various denoising timesteps. As presented in Table 19, our full method ("Learned energy") achieves a significantly higher and more positive correlation compared to both the external verifier and the baseline energy model without MRNCL. Notably, in the critical intermediate stages of denoising ($t = 0$ to $t = 7$), our method maintains a strong positive correlation ($> 0.4$), indicating that the energy landscape reliably guides the search towards better solutions globally.

Table 19: Kendall-$\tau$ rank correlation (mean and std) between the verifier score/energy and the ground-truth quality across denoising steps $t$. Our intrinsic energy verifier demonstrates superior global alignment compared to an external verifier and the baseline (w/o MRNCL).

| Denoising Step $t$ | Learned External Verifier | Intrinsic Energy (w/o MRNCL) | Intrinsic Energy (Ours) |
|---|---|---|---|
| 0 | -0.0978 (0.1750) | 0.2300 (0.0854) | **0.4760 (0.1492)** |
| 1 | -0.0976 (0.1750) | 0.2300 (0.0854) | **0.4744 (0.1490)** |
| 2 | -0.0980 (0.1756) | 0.2308 (0.0854) | **0.4760 (0.1492)** |
| 3 | -0.0910 (0.1834) | 0.2290 (0.0838) | **0.4642 (0.1362)** |
| 4 | -0.0824 (0.1678) | 0.2368 (0.0800) | **0.4094 (0.1538)** |
| 5 | -0.1292 (0.1718) | 0.2506 (0.0594) | **0.4414 (0.1300)** |
| 6 | -0.1318 (0.1748) | 0.2638 (0.0766) | **0.3978 (0.2466)** |
| 7 | -0.3618 (0.1446) | 0.2262 (0.1194) | **0.4508 (0.2024)** |
| 8 | -0.0316 (0.2854) | 0.3214 (0.2138) | -0.1570 (0.1852) |
| 9 | -0.2686 (0.0322) | 0.3712 (0.0520) | -0.3094 (0.0368) |

**Indirect Evidence: Search Success.** The most conclusive proof of global alignment is the final outcome. The fact that our framework successfully scales—improving the $15 \times 15$ Maze success rate from a baseline of 6.25% to 88.28% (Table 5)—serves as strong empirical evidence that a global change to the energy landscape has occurred. Such a dramatic improvement in global search performance would be unattainable if the energy landscape remained locally fragmented and globally unaligned.

## E EXTENSION OF hMCTS TO CONVENTIONAL DIFFUSION MODELS

The inference method hMCTS of VFScale is not limited to energy-based diffusion models. With an external verifier, it can be applied to conventional models that predict noise directly. We validated this in the Maze task by replacing the learned energy with MSE as the reward, confirming hMCTS's

compatibility. Its gains over best-of-N search from the Table 20 below further demonstrate broad applicability across diffusion models.

Table 20: Success rate on Maze (grid size 15) using a conventional noise-predicting diffusion model (DDPM) guided by an external ground-truth verifier. We compare the performance of hMCTS with BoN under different compute budgets ($N$). Here, the reward is defined as the negative MSE to the ground-truth solution. Results validate the compatibility of hMCTS with conventional diffusion and its improved efficiency over best-of-$N$ sampling.

| Methods | $N = 1$ | $N = 11$ | $N = 21$ | $N = 41$ | $N = 81$ | $N = 161$ |
|---|---|---|---|---|---|---|
| DDPM, hMCTS, Ground-truth guided | 0.0312 | 0.1328 | 0.1406 | 0.1875 | 0.2188 | 0.2422 |
| DDPM, BoN, Ground-truth guided | 0.0312 | 0.1094 | 0.1587 | 0.1746 | 0.2031 | 0.2422 |

## F   COMPUTATIONAL COST AND FAIR COMPARISON

This section provides a transparent analysis of the computational costs associated with the VFScale framework, covering both training and inference. We also detail our methodology for ensuring a fair comparison between different inference-time search strategies. All benchmarks were run on a machine with one 80GB VRAM GPU and 16 CPU cores.

### F.1   TRAINING COST

By design, VFScale tackles a more complex training objective than the baseline to shape a search-friendly energy landscape. This increased complexity results in a higher demand for training time and memory, as detailed in Table 21 and Table 22.

While the LRNCL and KL losses increase the training cost, this upfront investment is a crucial trade-off. For the original method, performance plateaus once the loss converges, regardless of additional training. In contrast, our approach enables the base model's performance to continuously improve with a larger training budget, which in turn unlocks significantly enhanced test-time scalability.

Table 21: Training time in hours for Sudoku and Maze task. The VFScale training objectives require more time but lead to models with superior scalability.

| Method | 100,000 Steps (hours) | 200,000 Steps (hours) |
|---|---|---|
| Original | 2 | 4 |
| +LRNCL | 4 | 5 |
| +KL | 3 | 6 |
| +LRNCL+KL | 6 | 7 |

Table 22: Peak GPU memory usage (VRAM in GB) during training with a batch size of 64.

| Method | Sudoku (GB) | Maze (GB) |
|---|---|---|
| Original | 3.28 | 3.40 |
| +LRNCL | 4.56 | 5.86 |
| +KL | 4.44 | 21.21 |
| +LRNCL+KL | 5.57 | 23.05 |

### F.2   INFERENCE COST AND FAIR COMPARISON

To ensure a fair and meaningful comparison between inference methods (BoN, MCTS, and our hMCTS), we allocate the **same Number of Function Evaluations (NFE)** to each method for a given problem instance. Since the forward pass through the model is the primary computational bottleneck, controlling for NFE allows us to isolate the efficiency of the search strategy itself.

Table 23 reports the wall-clock inference time. As expected, hMCTS exhibits a modest overhead (up to 31% longer than BoN at N=81). This is a known trade-off, as the sequential nature of tree expansion in MCTS-based methods is inherently less parallelizable than the embarrassingly parallel BoN approach. However, hMCTS is still significantly more efficient than a pure MCTS implementation.

Crucially, our central claim holds: for an identical computational budget (NFE), hMCTS delivers superior performance over both BoN and MCTS. The difference in wall-clock time stems from the search strategy's implementation rather than from the core computational complexity. We view this as a manageable engineering trade-off for the substantial performance gains achieved.

Table 23: Wall-clock inference time per sample (seconds) on the Maze task. For the same NFE, hMCTS provides superior results, justifying its modest time overhead compared to BoN.

| Method | N=1 | N=11 | N=21 | N=41 | N=81 |
|---|---|---|---|---|---|
| BoN | 2.17 | 2.80 | 5.63 | 11.01 | 20.16 |
| hMCTS (ours) | 1.02 | 3.28 | 6.56 | 12.66 | 26.48 |
| MCTS | 1.44 | 7.59 | 14.03 | 28.56 | 57.66 |

## G  BROADER IMPACTS

This paper aims to advance machine learning, particularly diffusion-based generative models. While our improvements promise higher-quality AI-generated content, they also carry potential societal risks. Accordingly, we must remain vigilant to unintended negative outcomes and guard against any unethical or illicit applications of this technology.

## H  THE USE OF LLMS

In the preparation of this manuscript, Large Language Models (LLMs) were used as an assistive tool to improve the quality of the work. The specific uses include:

- **Writing and Language Refinement:** LLMs were utilized to paraphrase sentences for clarity, correct grammatical errors, and refine the overall language to meet high academic standards.

- **Coding and Engineering Assistance:** LLMs were employed to generate boilerplate code, assist in debugging, and accelerate the implementation of standard algorithms, which supported the engineering of our experimental framework.

The authors reviewed, edited, and validated all LLM-generated content (both text and code) to ensure its technical accuracy and originality. The authors take full responsibility for all content presented in this paper.

## I  LIMITATIONS AND FUTURE WORK

Our inference framework primarily relies on MCTS, which presents two key limitations: (1) limited compatibility with parallel computing, and (2) challenges in effectively evaluating node quality during the early stages of denoising. Future work could explore integrating alternative search strategies, such as those proposed by Wu et al. (2024). Additionally, to enhance performance-energy consistency, we introduce linear-regression negative contrastive learning, which enforces a linear relationship between energy and the distance to real samples. Further investigation is needed to assess the broader implications of this constraint and explore alternative regularization approaches. Lastly, while our current implementation utilizes Gaussian noise for branching, other diffusion-based branching mechanisms remain an open area for exploration.

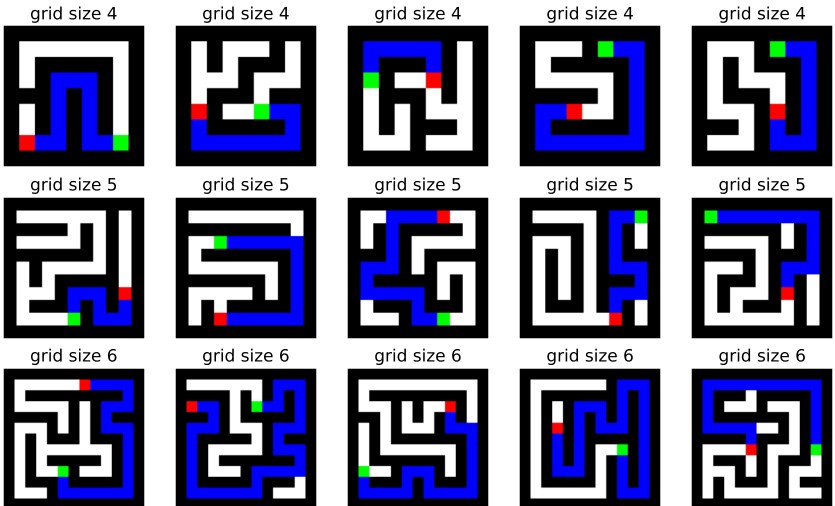

Figure 9: Visualization of training maze dataset.

## J    VISUALIZATION OF RESULTS

### J.1    VISUALIZATION OF MAZE EXPERIMENTS

This section presents visualizations of the training in Fig. 9, test Maze data in Fig. 10, and samples generated by different methods in Fig. 11. In the visuals, black pixels denote walls, green represents the starting point, red represents the goal point, blue marks the solved path, and white represents the feasible area. All visualizations are based on a few representative samples. The results from the training and test sets clearly show that the tasks in the test set are notably more challenging than those in the training set. Visual comparisons of samples generated by different methods reveal that the originally trained model, regardless of the inference strategy, performs consistently worse than VFScale.

### J.2    VISUALIZATION OF SUDOKU EXPERIMENTS

This section presents visualizations of the training and test Sudoku data in Fig. 12, and representative samples generated by different methods in Fig. 13. In the visuals, black numbers denote the condition, green numbers represent correct predictions, and red numbers represent wrong predictions. All visualizations are derived from a few representative samples. The comparison between the training and test sets clearly indicates that the tasks in the test set are significantly more difficult than those in the training set. When comparing the samples generated by different methods, it is evident that the originally trained model, regardless of the inference strategy, consistently underperforms VFScale.

## K    ETHICS STATEMENT

All authors have read and adhered to the ICLR Code of Ethics. This research focuses on the foundational capabilities of generative models for algorithmic reasoning tasks. The datasets used are synthetically generated and do not contain personally identifiable information or sensitive data. Our work does not involve human subjects. While our primary contribution is algorithmic, we acknowledge that advancements in generative models can have broader societal impacts. We believe the potential for misuse of this specific technology is low, but we encourage the responsible development and application of all AI systems.

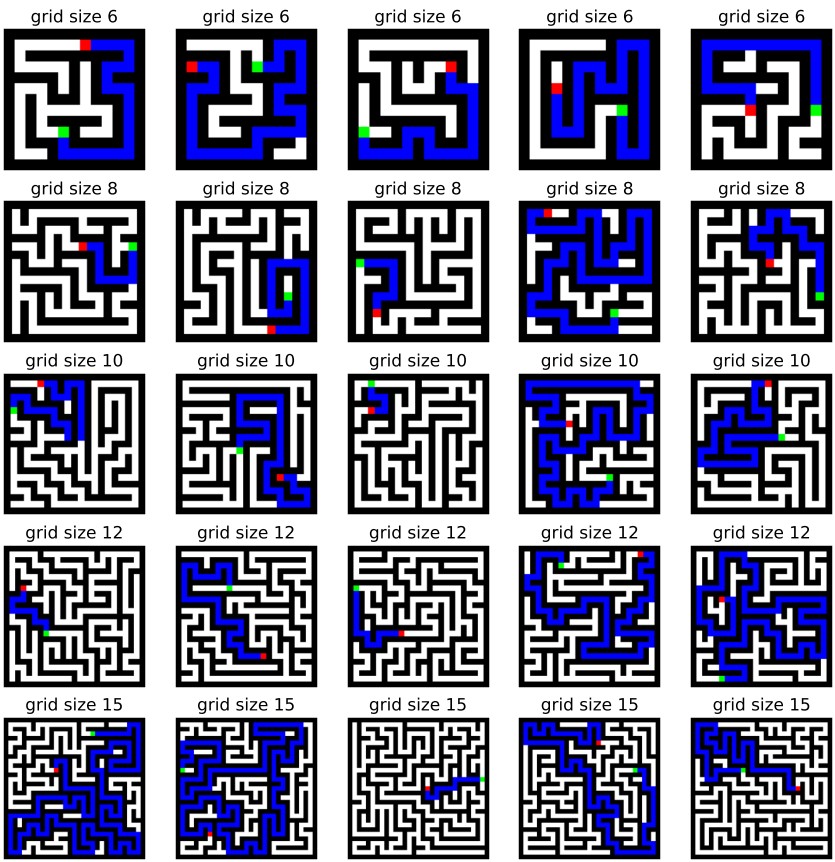

Figure 10: Visualization of test maze dataset, where the blue paths are ground-truth solutions.

## L    REPRODUCIBILITY STATEMENT

To ensure the reproducibility of our research, we have made the complete source code, datasets, and pre-trained model checkpoints publicly available at `https://github.com/AI4Science-WestlakeU/VFScale`. The repository contains the full implementation of our proposed VFScale framework, including all training objectives (e.g., MRNCL) and the hMCTS inference algorithm. Furthermore, comprehensive details regarding our experimental setup, model architectures, hyperparameter settings, and computational resources are provided in the Appendix.

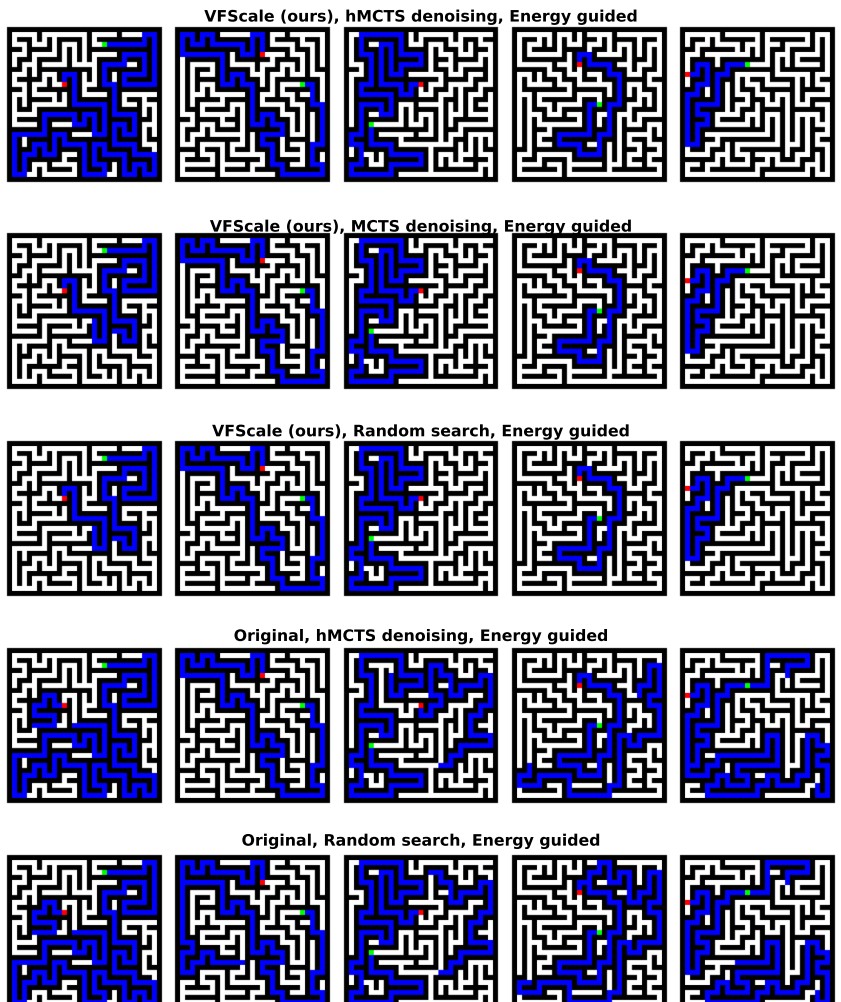

Figure 11: Visualization of samples generated by different training and inference methods.

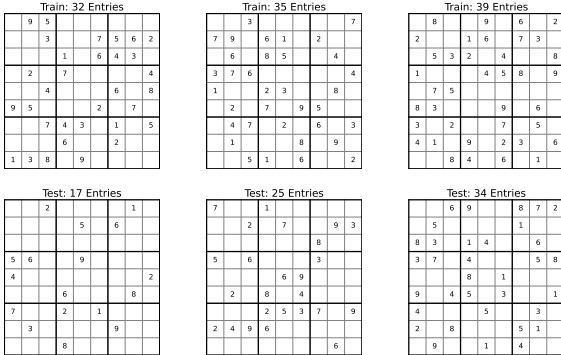

Figure 12: Visualization of training and test Sudoku dataset.

Figure 13: Visualization of samples generated by different training and inference methods.

