# OpenReview forum: "VFScale: Intrinsic Reasoning through Verifier-Free Test-time Scalable Diffusion Model"
_ICLR.cc/2026/Conference — ICLR 2026 Poster_

### Official Review · Reviewer_FVwD · 2025-10-28

**Soundness:** 2
**Presentation:** 2
**Contribution:** 2
**Rating:** 4
**Confidence:** 4

**Summary:**

This paper aims to address the lack of test-time scaling capabilities in diffusion models for complex reasoning tasks (e.g., maze solving and Sudoku). The core idea of this work is to use the diffusion model's own intrinsic energy function as a verifier to guide the search.

To achieve this, VFScale incorporates two key training innovations: Monotonic-Regression Negative Contrastive Learning (MRNCL) and KL regularization—to improve the energy landscape so that the model’s learned energy can act as an intrinsic verifier. At inference time, a hybrid Monte Carlo Tree Search (hMCTS) algorithm is proposed. This algorithm uses Best-of-N (BoN) for broad exploration during the early, noisy denoising stages and switches to MCTS for deep exploitation in the later, less-noisy stages.

Experimental results show that VFScale (e.g., trained on 6x6 mazes) can achieve a high success rate on complex tasks far beyond the training distribution (e.g., 15x15 mazes), where baseline models fail completely.

**Strengths:**

1. **Interesting Problem Setup:**
   The paper tackles the idea of *intrinsic reasoning*—removing external verifiers for diffusion-based reasoning—which is conceptually appealing and timely. This aligns well with the emerging research trend on *test-time scaling* and reasoning within generative models.

2. **Clear Positioning within Existing Literature:**
   The paper establishes clear connections to prior work, particularly Du et al. (2022, 2024) on energy-based diffusion, effectively situating the contribution within the broader context of diffusion-driven reasoning research.

3. **Algorithmic Insight of hMCTS:**
   The hybrid design of hMCTS demonstrates sound algorithmic intuition. Employing Best-of-N (BoN) for parallel exploration during early noisy stages and transitioning to MCTS for fine-grained exploitation in later denoising stages is well-motivated and consistent with the diffusion dynamics.

4. **Strong OOD Generalization Performance:**
   The experiments show notable improvements in out-of-distribution (OOD) generalization, suggesting that the proposed intrinsic reasoning framework may enhance robustness beyond the training distribution.

**Weaknesses:**

1. **Marginal Gains from MRNCL:**
   The improvement attributed to MRNCL appears small and inconsistent across tasks. The MRNCL objective, which enforces a linear energy–distance relationship, feels somewhat ad hoc. Imposing strict linearity between the $L_2$ distance and energy may oversimplify the energy landscape and lacks clear theoretical justification. Empirically, much of the observed gain seems to stem from the KL term rather than MRNCL itself (see Tables 1 and 3, particularly in the Sudoku experiments).

2. **Questionable Inference Design in Continuous Space:**
   The proposed hMCTS in practice appears to reduce to a Best-of-N sampling strategy with branching Gaussian perturbations. It remains unclear what specific advantages MCTS provides in this continuous setting, as its benefits are typically most pronounced in discrete action spaces.

3. **Unclear Experimental Setup and Visualization:**
   The implementation details for the Maze task are under-specified. While the paper implies continuous-valued diffusion states, the Maze itself is inherently discrete. It is unclear how intermediate diffusion states are visualized (e.g., Fig. 1) or discretized back into paths. This ambiguity raises questions about whether the claimed “continuous reasoning formulation” genuinely applies to these discrete tasks.

4. **Limited Novelty of Components:**
   The major components—contrastive energy training, KL regularization, and MCTS-based inference—are largely adapted from existing literature (e.g., Du et al., 2024; Silver et al., 2016). The central contribution, MRNCL, introduces a monotonic regression term that offers modest empirical improvement, making the overall technical novelty somewhat incremental.

5. **Limited Evaluation Scope:**
   The experiments are confined to synthetic reasoning tasks (Maze and Sudoku). While suitable for controlled evaluation, the absence of results on larger or real-world structured reasoning benchmarks limits the generality and impact of the work.

---

#### Miscellaneous Issue

1. **Inconsistent Notation:**
   The definition of the energy function is inconsistent across sections. In Du et al. (2024) and line 198, it is defined with both the input and its corresponding solution, whereas in other parts of the paper, the energy function is presented as depending solely on the solution.

**Questions:**

1. **Computation of $p_{\theta, t}(x)$:**
   How is $p_{\theta, t}(x)$ computed exactly? Please also clarify its relationship to the predicted energy function.

2. **Representation of Discrete Data:**
   Please elaborate on how the Maze and Sudoku datasets are represented. Specifically, how are discrete states encoded into continuous vectors for diffusion modeling and energy computation?How are the intermediate Maze visualizations (e.g., Fig. 1) generated? Are they obtained by projecting continuous latent states, or by decoding discrete states from intermediate diffusion steps?

4. **Energy–Distance Relationship:**
   What is the theoretical motivation for enforcing a *linear* relationship between the predicted energy and the \( L_2 \) distance? Would a monotonic but non-linear mapping (e.g., logistic or exponential) be equally valid or even more flexible?

5. **MCTS vs. Best-of-N Sampling:**
   Since each MCTS expansion involves multiple Gaussian perturbations, what makes your MCTS procedure fundamentally different from a multi-step Best-of-N or beam search strategy?

---

#### Miscellaneous

1. **Definition of “Guided with Ground Truth” (Table 2):**
   The term *“guided with ground truth”* in Table 2 is not clearly defined. Please clarify the exact experimental setup and what this guidance entails.

---

> ### Author Response · Authors · 2025-11-18
> **Official Response to Reviewer FVwD (1)**
>
> We sincerely thank Reviewer FVwD for their thorough and in-depth review of our work. We have carefully addressed all raised concerns and provide our detailed, point-by-point responses below. A revised manuscript has been uploaded, with all modifications and new sections clearly marked in **blue**  for easy tracking.
>
> >Q1: The improvement attributed to MRNCL appears small and inconsistent across tasks. The MRNCL objective, which enforces a linear energy–distance relationship, feels somewhat ad hoc. Imposing strict linearity between the distance and energy may oversimplify the energy landscape and lacks clear theoretical justification. Empirically, much of the observed gain seems to stem from the KL term rather than MRNCL itself (see Tables 1 and 3, particularly in the Sudoku experiments).
>
> We would like to clarify the distinct and **complementary roles** of $L_{MRNCL}$ and $L_{KL}$, as their contributions are **synergistic, not marginal.**
>
> **Complementary Roles: Reliability (MRNCL) vs. Smoothness (KL):**
>
> MRNCL and $L_{KL}$ solve two distinct, orthogonal problems. $L_{KL}$ provides **smoothness**, making the search process easier while $L_{MRNCL}$ provides **reliability**, ensuring that lower energy scores *reliably* correspond to higher-quality solutions.
>
> This reliability is a **precondition** for our intrinsic verifier to work. The KL loss alone does not enforce this. As evidence, **Table 4** shows that MRNCL *alone* (w/o KL) boosts performance-energy consistency from ~73% to over **84%**.
>
> **Explaining the Maze vs. Sudoku Discrepancy:**
>
> The reviewer correctly observes different effects on Maze and Sudoku in Table 3. This does not show MRNCL is weak; it reveals the different bottlenecks of the baseline models:
>
> **For Maze:**  The baseline ("Original, BoN," Table 3) **completely fails** (10.94% at N=161). This indicates its landscape is *both* non-smooth and unreliable. This is precisely why Maze **requires the synergy of both losses:** the Full Model (70.31%) significantly outperforms both "w/o KL" (62.50%) and "w/o MRNCL" (60.94%).
>
> **For Sudoku:** The baseline ("Original, BoN," Table 3) is **partially effective** (28.12% at N=321). This suggests its primary bottleneck is poor landscape *smoothness*. Consequently, $L_{KL}$ provides the largest single boost ("w/o MRNCL" reaches 41.41%).
>
> This fully optimized landscape (from both losses) is what enables our hMCTS search to achieve the final, superior system performance shown in Table 5.
>
> **Justification for the Linear Constraint (LRNCL):**
>
> Regarding the "ad hoc" criticism: our motivation for a linear constraint was **simplicity**, but its selection was **empirically validated, not arbitrary.** We *explicitly tested this assumption* against other constraints in **Appendix D.2 (Table 12) in the original submission.**
>
> Copied table  from Appendix D.2 (Table 12)
>
> ||Success rate on Maze|||||
> |-----------------------|--------|--------|--------|--------|---------|
> | Training Methods|N=1|N=11|N=21|N=41|N=81|
> | Original, BoN,energy guided|0.0625|0.0469|0.0547|0.0781|0.1016|
> | +LRNCL, BoN, energy guided|0.2812|**0.4688**|**0.4922**|**0.5547**|**0.5859**|
> | +QRNCL, BoN, energy guided|**0.3516**|0.3672|0.3906|0.3984|0.3828|
> | +RBNCL, BoN, energy guided|0.1875|0.4219|0.4766|0.5234|0.5781|
> | +FTNCL, BoN, energy guided|0.3125|0.4141|0.4844|0.4922|0.5156|
>
> Notably, an interesting trade-off emerges as Table 12 shows: QRNCL has the best initial performance but scales poorly. Conversely, RBNCL and LRNCL show superior scalability, with the simpler LRNCL being the most effective. This demonstrates that LRNCL's linear assumption, chosen for its simplicity, provides the best balance for calibrating the energy landscape to improve **test-time scalability.**
>
> In summary, MRNCL is (1) a necessary **reliability** component, (2) **synergistic** with KL, succeeding in both tasks by enabling superior test-time scaling (Table 5), and (3) **empirically justified** as its linear form provides the best scalability (Table 12).

---

> > ### Author Response · Authors · 2025-11-18
> > **Official Response to Reviewer (2)**
> >
> > >Q2: Questionable Inference Design in Continuous Space：**The proposed hMCTS in practice appears to reduce to a Best-of-N sampling strategy with branching Gaussian perturbations. It remains unclear what specific advantages MCTS provides in this continuous setting, as its benefits are typically most pronounced in discrete action spaces.
> >
> > While MCTS is often associated with discrete domains, its core principles—value-based search and the exploration/exploitation trade-off (e.g., [1, 2])—are general. Our hMCTS implementation is **mechanistically distinct** from "BoN with branching" by retaining two foundational MCTS mechanisms that BoN lacks:
> >
> > 1. **UCB-based Selection:** BoN is a parallel, 'one-shot' search. Our MCTS phase builds a tree and uses a UCB policy (Eq. 7) to ***selectively** allocate more compute* to promising branches, formally balancing exploitation (low energy) and exploration (high uncertainty).
> > 2. **Value Backpropagation:** The value (energy) from a simulation is backpropagated (Algorithm 1, Line 13) to update the $Q$-values of all parent nodes in that path. This allows the tree to learn from its rollouts—a crucial learning mechanism that BoN lacks.
> >
> > The reviewer is correct that our *expansion step* (Eq. 8) uses Gaussian perturbations; this is a **principled adaptation** to the continuous, stochastic state space of diffusion models. The critical point is that the *search itself* is **not random but is actively guided** by the UCB and value backpropagation mechanisms.
> >
> > This mechanistic difference is not merely theoretical; it translates directly to a large empirical gain. Our ablation study on the MCTS start step `$t_s$` (Appendix D.3, Figure 8 & Table 14) shows this clearly:
> >
> > 1. The pure BoN strategy (VFScale tr. at `$t_s=0$`) achieves **0.6562** success.
> > 2. Our optimal hybrid hMCTS (peaking at `$t_s=4$`) achieves **0.7969** success.
> >
> > This **~14% absolute improvement** is the direct result of the MCTS phase's guided, value-driven search, confirming the MCTS component is critical and not a reduction to BoN.
> >
> > [1] Couëtoux, Adrien, Levente Kocsis, and Olivier Teytaud. "Continuous upper confidence trees." *Learning and Intelligent Optimization*. Springer, 2011.
> >
> > [2] Wang, Chao, and Risto Miikkulainen. "Monte Carlo tree search for continuous action spaces in POMDPs." *2014 IEEE Conference on Computational Intelligence and Games*. IEEE, 2014.
> >
> > >Q3: The implementation details for the Maze task are under-specified. While the manuscript implies continuous-valued diffusion states, the Maze itself is inherently discrete. This ambiguity raises questions about whether the claimed “continuous reasoning formulation” genuinely applies to these discrete tasks.
> >
> > We thank the reviewer for highlighting this omission in our implementation details. The reviewer is correct that the task is discrete and our states are continuous. This discrete-continuous-discrete pipeline, which is a common approach for applying generative models to discrete data [1], allows the model to leverage gradient-based refinement.
> >
> > Our "continuous reasoning formulation" is robust and applies as follows:
> >
> > **Discrete $\rightarrow$ Continuous Encoding:**  The $H \times W$ discrete Maze task is first encoded into a continuous tensor. We treat each grid cell as having one of $C$ states (e.g., "path," "wall," "empty") and use a one-hot encoding for each cell. This transforms the discrete $H \times W$ grid into a continuous $H \times W \times C$ tensor, $x_0$, which serves as the model's input.
> >
> > **Continuous Diffusion Process:**  Our energy function, $E_\theta$, and all search algorithms (BoN/hMCTS) operate *entirely* in this continuous $H \times W \times C$ space. All diffusion, noise prediction, and energy calculations are performed on these continuous-valued tensors.
> >
> > **Decoding to Discrete:**  To obtain the final discrete solution, we apply a standard **`argmax`** operation along the channel dimension ($C$) of the final output tensor $\hat{x}_0$. This collapses the probability distribution back to the single most likely discrete state, and this final discrete grid is evaluated for success .
> >
> > We have added this process to **Appendix B.6** of the revised manuscript.
> >
> > [1] Du, Yilun, Jiayuan Mao, and Joshua B. Tenenbaum. "Learning Iterative Reasoning through Energy Diffusion." *International Conference on Machine Learning*. PMLR, 2024.

---

> > > ### Author Response · Authors · 2025-11-18
> > > **Official Response to Reviewer (3)**
> > >
> > > >Q4: How are the intermediate Maze visualizations (e.g., Fig. 1) generated?
> > >
> > > We should clarify that the visualizations in Figure 1 (and throughout the paper) depict the **final, discrete solution paths** generated after the *full* denoising process, not the intermediate, continuous states.
> > >
> > > The process begins with a continuous Gaussian noise tensor, $x_T$ (with shape $H \times W \times 2$). We then apply our iterative denoising sampling process to generate the solution's one-hot encoding, $x_0$ (also of shape $H \times W \times 2$).
> > >
> > > Subsequently, an `argmax` operation is applied to the $x_0$ tensor to obtain the final discrete solution map (of shape $H \times W$). This map indicates whether each grid cell is part of the solution path.
> > >
> > > To render the final visualization, we first draw the base Maze map based on its layout data: walls are colored **black**, and empty spaces are colored **white**. Then, the start and end points are marked as **red** and **green**, respectively. Finally, the solution path derived from the `argmax` operation is then overlaid onto this map and colored **blue**.
> > >
> > > We have included the visualization process in **Appendix B.6** of the revised manuscript.
> > >
> > > >Q5: Limited Novelty of Components:** The major components—contrastive energy training, KL regularization, and MCTS-based inference—are largely adapted from existing literature (e.g., Du et al., 2024; Silver et al., 2016). The central contribution, MRNCL, introduces a monotonic regression term that offers modest empirical improvement, making the overall technical novelty somewhat incremental.
> > >
> > > We respectfully clarify that our contribution is not an incremental application of existing components, but rather a novel framework that diagnoses and solves a **fundamental bottleneck** in diffusion-based reasoning. Our central goal is to achieve **scalable intrinsic reasoning**  by enabling diffusion models to scale their compute at **test-time *without* an external verifier** .
> > >
> > > While Du et al. (2024) provides a strong baseline, we identify *why* it fails to generalize: its intrinsic energy function is a poor verifier for scaling. As shown in Table 3, the baseline ("Original, BoN") shows marginal gains from scaling, with success on the 15x15 Maze failing to exceed 11%. Our analysis is **the first** to pinpoint this failure to a lack of **performance-energy consistency**.
> > >
> > > The MRNCL loss is not a general-purpose regularizer; it is a *targeted* solution to this specific **consistency** problem. As shown in Table 4, MRNCL alone improves this consistency metric by over 10 percentage points (from ~73% to ~84%). This enhanced consistency unlocks the model's true potential. The "modest empirical improvement" characterization overlooks the final result: our full framework (VFScale tr. + hMCTS) achieves an **88% success rate** on the 15x15 Maze. This is a transformative **leap from the ~11% baseline**, demonstrating that our contributions successfully solve the scaling problem where prior work fails.
> > >
> > > Furthermore, our **hybrid MCTS (hMCTS)** is not a simple application of MCTS. It is a novel inference algorithm designed for diffusion. It strategically **combines** the broad, **parallel exploration** of Best-of-N (BoN) in the noisy initial stages with the deep, **adaptive exploitation** of MCTS in later stages, achieving a more effective exploration-exploitation balance.
> > >
> > > In summary, our novelty lies in (1) identifying the **"performance-energy consistency" bottleneck** of intrinsic verifiers, (2) proposing the targeted MRNCL loss to fix it, and (3) developing the synergistic hMCTS algorithm to exploit this improved energy landscape **for significant scalability.**

---

> > > > ### Author Response · Authors · 2025-11-18
> > > > **Official Response to Reviewer (4)**
> > > >
> > > > >Q6: Limited Evaluation Scope, more experimental scenarios.
> > > >
> > > > We agree that extending to non-grid domains is a critical frontier. However, for this foundational study, focusing on grid-based tasks (Maze and Sudoku) was a **deliberate methodological choice** to ensure precise experimental control.
> > > >
> > > > This setting allow us to define distinct "complexity gradients"—such as scaling Maze geometry ($6\times6 \to 15\times15$) or increasing Sudoku sparsity—to rigorously quantify the generalization gap, which is often ambiguous in unstructured tasks. Furthermore, the continuous nature of these grids enabled us to **explicitly visualize the energy landscape (Fig. 4)**, providing mechanistic proof that our MRNCL objective effectively smooths the optimization surface. Adhering to these benchmarks also ensures a strict, fair comparison with the primary baseline, confirming that the dramatic improvements observed (e.g., **88% vs. 0%** on large Mazes) are genuine algorithmic breakthroughs rather than artifacts of domain shifting.
> > > >
> > > > Crucially, **there are no structural barriers** preventing VFScale from extending to non-grid CSPs. Our framework is mathematically agnostic to data topology: the MRNCL objective operates on scalar energy values, and the hMCTS engine navigates the landscape independent of the input structure. For discrete problems like Graph Coloring, the $L_2$ distance in our objective can be naturally substituted with **Hamming distance** or **constraint violation counts** without altering the core logic. Therefore, having established the theoretical validity of verifier-free scaling in this controlled environment, we view the extension to unstructured domains as a promising and straightforward avenue for future work.
> > > >
> > > > [1] Du, Yilun, Jiayuan Mao, and Joshua B. Tenenbaum. "Learning Iterative Reasoning through Energy Diffusion." *International Conference on Machine Learning*. PMLR, 2024.
> > > >
> > > > >Q7: Inconsistent Notation:**The definition of the energy function is inconsistent across sections. In Du et al. (2024) and line 198, it is defined with both the input and its corresponding solution, whereas in other parts of the manuscript, the energy function is presented as depending solely on the solution.
> > > >
> > > > The reviewer is correct: the energy function is always conditioned on the input $c$ (formally introduced as $E_{\theta}(c, x_t, t)$ in Sec. 4.1). As stated in Footnote 3 on page 4, we omitted $c$ in subsequent notations purely for brevity (i.e., $E_{\theta}(x_t, t)$ is always implicitly $E_{\theta}(c, x_t, t)$.
> > > >
> > > > We agree that placing this convention in a footnote was suboptimal. To ensure clarity, we have moved this statement from the footnote directly into the main text of **Section 4.1** in the revised manuscript.
> > > >
> > > > >Q8: Computation of $p_{\theta,t}(x)$: How is $p_{\theta,t}(x)$computed exactly? Please also clarify its relationship to the predicted energy function.
> > > >
> > > > We thank the reviewer for this question, which allows us to clarify the computation of the $L_{KL}$ term.
> > > >
> > > > **Relationship between $p_{\theta,t}(x)$ and Energy :**
> > > >
> > > > The term $p_{\theta,t}(x)$ represents the probability distribution of the sample at denoising step $t$, as parameterized by our model. This distribution is implicitly defined by our learned energy function $E_{\theta}(x_t, t)$ via the standard energy-based model formulation (i.e., a Boltzmann distribution):$p_{\theta,t}(x) \propto \exp(-E_{\theta}(x_t, t))$. Therefore, we do not compute the probability density function $p_{\theta,t}(x)$ explicitly, as this would require computing the intractable partition function. Instead, our $L_{KL}$ loss is formulated to only require sampling from this distribution and estimating its entropy.
> > > >
> > > > **Computation of $L_{KL}$:**
> > > >
> > > > $$L_{KL}=E_{t,p_{\theta,t}(x)}[E_{stop-grad(\theta)}(x)]+E_{t,p_{\theta,t}(x)}[log~p_{\theta,t}(x)]$$
> > > >
> > > > This expectation is computed by first sampling $x_t \sim p_{\theta,t}(x)$, which is achieved by running the denoising (sampling) process. The first term, $E_{t,p_{\theta,t}(x)}[E_{stop-grad(\theta)}(x)]$, is estimated by calculating the average energy $E_{\theta}(x_t, t)$ of these generated samples. The second term, $E_{t,p_{\theta,t}(x)}[log~p_{\theta,t}(x)]$, is the negative differential entropy of the distribution $p_{\theta,t}(x)$. As stated in the paper, we estimate this entropy term non-parametrically from the generated samples using the k-nearest-neighbor (k-NN) entropy estimator [1].
> > > >
> > > > [1] Lombardi, Damiano, and Sanjay Pant. "Nonparametric k-nearest-neighbor entropy estimator." *Physical Review E* 93.1 (2016): 013310.

---

> > > > > ### Author Response · Authors · 2025-11-18
> > > > > **Official Response to Reviewer (5)**
> > > > >
> > > > > >Q9: Energy–Distance Relationship:** What is the theoretical motivation for enforcing a *linear* relationship between the predicted energy and the ( L_2 ) distance? Would a monotonic but non-linear mapping (e.g., logistic or exponential) be equally valid or even more flexible?
> > > > >
> > > > > We agree that a formal proof for why LRNCL improves the energy landscape is a challenging but important direction. While our paper focuses on the empirical validation of VFScale, we can offer a **theoretical intuition**  grounded in established machine learning principles.
> > > > >
> > > > > Our LRNCL method extends standard contrastive learning. Whereas standard contrastive loss merely distinguishes between positive and negative samples—**treating all negatives equally**—MRNCL imposes a **metric structure**  on the energy landscape. Specifically, by regressing the energy of negative samples to their L2 distance from the ground-truth solution, MRNCL ensures the energy value reflects not just ***if***  a sample is wrong, but ***how wrong*** it is. The resulting predictable energy gradient makes the landscape significantly **easier for any search algorithm to navigate.**
> > > > >
> > > > > To validate this design choice, we **already conducted a comprehensive ablation study in Appendix D.2 (Table 12) in the original submission**. We directly compared: **LRNCL (Ours):** A linear constraint, **QRNCL (Quadratic):** A non-linear quadratic constraint, **RBNCL (Rank-based):** A rank-correlation-based loss, which is precisely the type of listwise/ranking loss.
> > > > >
> > > > >
> > > > > ||Success rate on Maze|||||
> > > > > |-----------------------|--------|--------|--------|--------|---------|
> > > > > |Training Methods|N=1|N=11|N=21|N=41|N=81|
> > > > > |Original, BoN,energy guided|0.0625|0.0469|0.0547|0.0781|0.1016|
> > > > > |+LRNCL, BoN, energy guided|0.2812|**0.4688**|**0.4922**|**0.5547**|**0.5859**|
> > > > > |+QRNCL, BoN, energy guided|**0.3516**|0.3672|0.3906|0.3984|0.3828|
> > > > > |+RBNCL, BoN, energy guided|0.1875|0.4219|0.4766|0.5234|0.5781|
> > > > >
> > > > >
> > > > >
> > > > > The **empirical results**  of this experiment (Table 12) are clear: while other constraints (like QRNCL) perform better at N=1 (0.3516 vs 0.2812), our **LRNCL (Linear) provides the strongest and most consistent test-time scalability** (e.g., 0.5859 at N=81, far exceeding QRNCL's 0.3828).
> > > > >
> > > > > Therefore, while our method lacks a formal theoretical guarantee, our **empirical results strongly justify** that LRNCL is a superior and rational design *for our specific goal* of scalability.
> > > > >
> > > > > >Q10: MCTS vs. Best-of-N Sampling:** Since each MCTS expansion involves multiple Gaussian perturbations, what makes your MCTS procedure fundamentally different from a multi-step Best-of-N or beam search strategy?
> > > > >
> > > > > The fundamental difference lies in the **computational budget allocation**. Best-of-N (BoN) and beam search are uniform or greedy strategies, whereas our MCTS is **adaptive**.
> > > > >
> > > > > A multi-step **BoN** would run $N$ independent trajectories in parallel and select the best one at the very end. It does not use intermediate results to re-allocate compute. A **beam search** greedily prunes paths at *each* step, keeping only the top-k candidates and permanently discarding all others.
> > > > >
> > > > > **Our MCTS Procedure** is an *adaptive* search strategy. The key difference comes from the **Selection** and **Backpropagation** steps: **(1)UCB Selection:**  We use the UCB policy to non-uniformly select which node to expand. This policy explicitly balances **exploitation** (revisiting promising branches that have low energy, i.e., high $Q$ values) and **exploration** (exploring less-visited branches). **(2) Backpropagation:**  After a simulation (rollout), the final energy of the solution is backpropagated to update the $Q$ values of all nodes along that path.
> > > > >
> > > > > This mechanism allows MCTS to **dynamically** allocate more computational budget (i.e., more rollouts) to branches that appear more promising as the search progresses, rather than treating all $N$ branches **equally**.
> > > > >
> > > > > This is precisely why we developed the **hybrid MCTS (hMCTS).** We leverage BoN in the noisy initial stages for broad, parallel **exploration**. Then, as the signal becomes **clearer**, we switch to MCTS to perform deep, adaptive **exploitation** on the most viable candidates, which is a more efficient use of computation than a simple BoN.
> > > > >
> > > > > >Q11: Definition of “Guided with Ground Truth” (Table 2).
> > > > >
> > > > > The term **guided with ground truth**  refers to the element accuracy w.r.t. the true solution, an **oracle guidance**  used to isolate and evaluate the maximum potential of the model's sample generation capability, separate from the quality of its learned energy function.

---

### Official Review · Reviewer_g4A4 · 2025-10-29

**Soundness:** 3
**Presentation:** 3
**Contribution:** 2
**Rating:** 6
**Confidence:** 2

**Summary:**

The paper proposes VFScale, a way to make diffusion models solve reasoning problems better without relying on an external verifier. It treats the model’s own energy as an intrinsic verifier and improves both training and inference: on the training side, a loss called MRNCL enforces that samples farther from the ground truth have monotonically higher energy and adds a KL term to smooth the landscape so lower energy better aligns with correctness; on the inference side, a hybrid search hMCTS uses Best-of-N early (broad exploration) and MCTS later (deep exploitation) to convert extra compute into higher success more efficiently.

**Strengths:**

- Treats diffusion energy as an intrinsic verifier, removing dependence on external verifiers and focusing on verifier accuracy + search efficiency.
- Training (MRNCL + KL) aligns low energy with correctness; inference (hMCTS) sensibly combines early Best-of-N with later MCTS to turn extra compute into better.

**Weaknesses:**

- Using diffusion energy as an intrinsic verifier and aligning it via MRNCL + KL is largely heuristic; the paper provides no formal results
- MRNCL relies on how “near/far” negatives are defined and enforces only local monotonicity; the paper does not analyze robustness to the distance metric nor show that local constraints imply a global ranking alignment.

**Questions:**

1. Have the authors tried using majority voting for the task? If the diffusion model can consistently output a path more often than the other, it can also be used as a verifier.
2. How about pitting MRNCL against margin / listwise ranking losses—and showing how results shift when you swap the “near/far” distance metric for negatives?
3. Any quick Kendall $\tau$ / Spearman $\rho$ between energy and correctness across timesteps to show the intrinsic-verifier story holds up?

---

> ### Author Response · Authors · 2025-11-18
> **Official Response to Reviewer g4A4 (1)**
>
> We thank Reviewer g4A4 for the valuable suggestions. Based on this feedback, we have conducted several new experiments and uploaded a revised version of our paper. We provide our detailed responses to each of reviewer's questions below, and all new content in the manuscript is marked in **blue** .
>
> >Q1: Have the authors tried using majority voting for the task? If the diffusion model can consistently output a path more often than the other, it can also be used as a verifier.
>
> Thank you for suggesting this important baseline. The premise that a consistent model should favor the correct solution is logical, so we **conducted this exact experiment**.
>
> We used our best-trained model ("VFScale tr. (ours)") on the 15x15 Maze task and generated N samples. Instead of using our intrinsic energy verifier, we selected the final solution via "Majority Voting" (i.e., the path that appeared most frequently).
>
>
> Success rate of different approaches on Maze with grid size 15 across different inference budget.
> |Methods|N=1|N=11|N=21|N=41|N=81|N=161|
> |:---|:---:|:---:|:---:|:---:|:---:|:---:|
> |BoN|0.2656|0.5859|0.6406|0.6875|0.6562|0.7031|
> |hMCTS denoising (ours)|**0.2656**|**0.6406**|**0.7266**|**0.7969**|**0.8203**|**0.8828**|
> |Majority voting|0.2656|0.2812|0.3047|0.3047|0.2969|0.2812|
>
>
>
> The results are striking and directly refute the reviewer's working premise for this problem. Majority Voting **completely fails to scale.**  Its performance peaks at N=11 (30.5%) and then *degrades* as the compute budget (N) increases. This experiment provides powerful evidence for the *type* of problem we are solving. The task of finding a correct path in a 15x15 OOD maze is a high-cardinality problem with an extremely sparse reward signal.
>
> The model does *not* produce "the correct path" and "a few common incorrect paths." Instead, it produces: **a few instances of the correct path**  (which our N=1 success rate of ~28% shows) and **vast "long tail" of *many thousands of unique incorrect paths*** . In this scenario, a passive consensus mechanism like Majority Voting fails. It is more likely to find a "fluke consensus" (e.g., 3 samples of "Incorrect Path A") than it is to find the correct path, which may only appear 1 or 2 times in a batch. The "majority" vote is almost always cast for an *incorrect* solution.
>
> **This experiment perfectly highlights *why* our framework is necessary.**  It proves that a passive, consensus-based verifier is insufficient for complex reasoning tasks. It demonstrates the need for an **active, guided search**—which is exactly what our framework provides. Our method uses the intrinsic energy function, shaped by MRNCL , as an *active co-pilot* to guide the hMCTS search toward the single **"needle in a haystack"** (the low-energy, correct solution).
>
> In conclusion, we are grateful for this suggestion, as the resulting experiment strongly reinforces our manuscript's central claim: successfully scaling on complex OOD reasoning tasks requires *both* (1) a well-shaped, scalable energy landscape and (2) an efficient, guided search algorithm (like hMCTS) to exploit it.

---

> ### Author Response · Authors · 2025-11-18
> **Official Response to Reviewer g4A4 (2)**
>
> >Q2: MRNCL relies on how “near/far” negatives are defined and enforces only local monotonicity; the paper does not analyze robustness to the distance metric nor show that local constraints imply a global ranking alignment.
>
> The reviewer correctly identifies that our constraint is applied *locally*, and it is fair to ask how this achieves a *global* effect.
>
> We confirm that we analyzed the robustness of our design choices in the appendix. We will address each of the reviewer's points using this data.
>
> **On "Robustness to the Distance Metric" (Appendix D.2, Table 12):**
>
> The reviewer is correct that our "linear" constraint (LRNCL) is one of many possible choices. To justify this choice, we explicitly analyzed its robustness in **Appendix D.2 in the original submission**  by comparing it against several other monotonic constraints, including a **quadratic (QRNCL).** The reviewer's suggestion of a "listwise ranking loss" is captured by our "RBNCL (Rank-based NCL)" baseline, which uses the rank correlation as a loss, without assuming a specific functional form.
>
>
> ||Success rate on Maze|||||
> |-------------------------------|----------------------|--------|--------|--------|---------|
> |Training Methods|N=1|N=11|N=21|N=41|N=81|
> |Original, BoN,energy guided|0.0625|0.0469|0.0547|0.0781|0.1016|
> |+LRNCL, BoN, energy guided|0.2812|0.4688|0.4922|0.5547|0.5859|
> |+QRNCL, BoN, energy guided|0.3516|0.3672|0.3906|0.3984|0.3828|
> |+RBNCL, BoN, energy guided|0.1875|0.4219|0.4766|0.5234|0.5781|
>
> This experiment reveals a crucial trade-off: QRNCL achieves the best $N=1$ performance but scales very poorly. In contrast, our LRNCL provides the best and most consistent **test-time scalability**, which is the primary goal of our work. This empirically justifies our choice of the linear constraint as the most robust *for our specific goal*.
>
> **On "How 'Near/Far' Negatives are Defined" (Appendix D.4, Table 17):**
>
> We compared our standard "Permutation-based Perturb" strategy with a more generic "Gaussian Noise Perturb" strategy. The results show that LRNCL **robustly** improves performance and unlocks test-time scalability *regardless* of the specific negative sample generation strategy employed. Both methods significantly outperform the "Original" baseline.
>
> **On "Local Constraints vs. Global Ranking Alignment" (Table 4 & New Experiment)**
>
> Our hypothesis is that by **repeatedly enforcing this simple, local, linear consistency across lots of samples, and diffusion timesteps during training**, we shape a *globally* more structured and reliable energy landscape.
>
> We have  **direct evidence** and **indirect evidence** to support this hypothesis: that a *global* change is occurring:
>
> - **Direct Evidence:** Our PEC metric, formally defined in Appendix A.1, is a **measure of global rank-order consistency** across many samples. As shown in **Table 4**, our training (e.g., "VFScale tr. w/o KL," which includes MRNCL) improves this global consistency metric from **~73% (Original) to ~84%** . This is a direct measurement of improved global alignment. We also conducted a new experiment measuring the **Kendall-$\tau$ rank correlation**—a standard global alignment metric—across denoising timesteps. The results (which we have added to Appendix D.6) confirm that our full training method ("VFScale tr.") achieves a **significantly higher and more stable global rank correlation** than the "Original" baseline.
>
>
> |$t$|Learned external verifier|Learned energy as intrinsic verifier w/o MRNCL|Learned energy as intrinsic verifier|
> |---------------------|---------------------------|------------------------------------------------|--------------------------------------|
> |0|-0.0978(0.1750)|0.2300(0.0854)|0.4760(0.1492)|
> |1|-0.0976(0.1750)|0.2300(0.0854)|0.4744(0.1490)|
> |2|-0.0980(0.1756)|0.2308(0.0854)|0.4760(0.1492)|
> |3|-0.0910(0.1834)|0.2290(0.0838)|0.4642(0.1362)|
> |4|-0.0824(0.1678)|0.2368(0.0800)|0.4094(0.1538)|
> |5|-0.1292(0.1718)|0.2506(0.0594)|0.4414(0.1300)|
> |6|-0.1318(0.1748)|0.2638(0.0766)|0.3978(0.2466)|
> |7|-0.3618(0.1446)|0.2262(0.1194)|0.4508(0.2024)|
> |8|-0.0316(0.2854)|0.3214(0.2138)|-0.1570(0.1852)|
> |9|-0.2686(0.0322)|0.3712(0.0520)|-0.3094(0.0368)|
>
>
>
> - **Indirect & Conclusive Evidence (Figure 3 & Table 5):** The most powerful evidence is the final outcome. The *very fact* that our full framework **successfully scales** —for instance, taking the 15x15 Maze success rate from a baseline of **6.25% to 88.28%** —is the proof that a *global* change to the energy landscape has occurred. Such improvement in search performance would be hard if the energy landscape remained unaligned and non-scalable.
>
> In summary, while our *mechanism* (LRNCL) is local, the *evidence* (Table 4, Table 5, Table 12, Table 17, and our new Kendall-$\tau$ analysis strongly indicates that it successfully achieves its *goal*: **creating a more robust, globally-aligned energy landscape that enables superior test-time scalability.**

---

> > ### Author Response · Authors · 2025-11-18
> > **Official Response to Reviewer g4A4 (3)**
> >
> > >Q3: Using diffusion energy as an intrinsic verifier and aligning it via MRNCL + KL is largely heuristic; the paper provides no formal results.
> >
> > This is a fair point. We clarify that while our framework is **empirically validated** rather than formally proven, the MRNCL constraint was not an *arbitrary* heuristic but a **deliberate design choice** .
> >
> > As the reviewer notes, formal guarantees for the scalability of such deep energy landscapes are often intractable. Therefore, our objective was to identify the specific inductive bias that best enables our paper's core goal: **test-time scalability**.
> >
> > To this end, we *already* included a comprehensive ablation study in **Appendix D.2 (Table 12)** in the original submission to validate this precise choice. We directly compared our LRNCL (Linear) against other plausible heuristics, including a non-linear quadratic constraint (QRNCL) and a rank-based loss (RBNCL).
> >
> >
> > ||Success rate on Maze|||||
> > |-----------------------|--------|--------|--------|--------|---------|
> > |Training Methods|N=1|N=11|N=21|N=41|N=81|
> > |Original, BoN,energy guided|0.0625|0.0469|0.0547|0.0781|0.1016|
> > |+LRNCL, BoN, energy guided|0.2812|**0.4688**|**0.4922**|**0.5547**|**0.5859**|
> > |+QRNCL, BoN, energy guided|**0.3516**|0.3672|0.3906|0.3984|0.3828|
> > |+RBNCL, BoN, energy guided|0.1875|0.4219|0.4766|0.5234|0.5781|
> >
> >
> >
> > The **empirical results**  of this experiment (Table 12) are clear: while other constraints (like QRNCL) perform better at N=1 (0.3516 vs 0.2812), our **LRNCL (Linear) provides the strongest and most consistent test-time scalability** (e.g., 0.5859 at N=81, far exceeding QRNCL's 0.3828).
> >
> > Therefore, while our method is indeed empirically driven and lacks a formal theoretical guarantee, our **empirical results (in Table 12)** strongly justify that LRNCL is a superior and well-validated design choice compared to other heuristics for our specific goal of scalability.
> >
> > >Q4: Any quick Kendall / Spearman between energy and correctness across timesteps to show the intrinsic-verifier story holds up?
> >
> > We have conducted this exact experiment to validate our "intrinsic verifier" story. We calculated the Kendall-τ rank correlation between the intrinsic energy and the true solution correctness (element-wise accuracy) across the denoising timesteps..
> >
> > During BoN inference with a compute budget of $K=40$, at each timestep $t_s$ (where larger $t_s$ implies a higher noise level), we estimate $\hat{x_0}$ for each branch. We then score these $\hat{x_0}$ candidates using two verifiers—the learned external verifier and the intrinsic verifier. We calculate the mean and standard deviation of the Kendall-$\tau$ correlation between each verifier's scores and the true element-wise accuracy, averaged across samples. A higher Kendall-$\tau$ indicates better consistency with the true element-wise accuracy.
> >
> >
> > |$t$|Learned external verifier|Learned energy as intrinsic verifier w/o MRNCL|Learned energy as intrinsic verifier|
> > |---------------------|---------------------------|------------------------------------------------|--------------------------------------|
> > |0|-0.0978(0.1750)|0.2300(0.0854)|0.4760(0.1492)|
> > |1|-0.0976(0.1750)|0.2300(0.0854)|0.4744(0.1490)|
> > |2|-0.0980(0.1756)|0.2308(0.0854)|0.4760(0.1492)|
> > |3|-0.0910(0.1834)|0.2290(0.0838)|0.4642(0.1362)|
> > |4|-0.0824(0.1678)|0.2368(0.0800)|0.4094(0.1538)|
> > |5|-0.1292(0.1718)|0.2506(0.0594)|0.4414(0.1300)|
> > |6|-0.1318(0.1748)|0.2638(0.0766)|0.3978(0.2466)|
> > |7|-0.3618(0.1446)|0.2262(0.1194)|0.4508(0.2024)|
> > |8|-0.0316(0.2854)|0.3214(0.2138)|-0.1570(0.1852)|
> > |9|-0.2686(0.0322)|0.3712(0.0520)|-0.3094(0.0368)|
> >
> >
> >
> > We have the following observations:
> >
> > Although the learned external verifier achieves a 0.99 correlation with the true element-wise accuracy, it still fails to provide fine-grained guidance for distinguishing between the **similar** samples in $\{x_0^i\}_{i=1}^{K=40}$. This further illustrates the necessity of the intrinsic verifier.
> >
> > Our training method significantly improves the consistency between the energy (intrinsic verifier) and the true element-wise accuracy during the late stages of denoising. It is worth noting that in the early denoising stages ($t = 8, 9$), the Kendall-$\tau$ is very low. This implies that **prematurely pruning** branches based on the intrinsic verifier would degrade the solution quality. **This observation also justifies our hMCTS method, which avoids branch pruning in these early stages.**
> >
> > Throughout the critical denoising stages (from $t=7$ down to $t=0$), our MRNCL-trained verifier shows a correlation that is approximately **2x higher**  than the baseline. This demonstrates that our method successfully shapes the energy landscape to "hold up" as a reliable verifier, which is precisely what enables our search algorithms (BoN and hMCTS) to scale effectively.

---

### Official Review · Reviewer_bGxA · 2025-10-31

**Soundness:** 3
**Presentation:** 3
**Contribution:** 3
**Rating:** 6
**Confidence:** 3

**Summary:**

This paper introduces VFScale, a framework for enhancing the test-time scalability of diffusion models for reasoning tasks, notably without requiring an external verifier. The core idea is to leverage the model's intrinsic energy function as a verifier. On the training side, the authors propose MRNCL loss and a KL regularization to improve the energy landscape. For inference, the authors adopt a hybrid MCTS for the denoising process to improve search efficiency. VFScale demonstrates good performance on challenging Maze and Sudoku tasks.

**Strengths:**

Strengths of this work are summarized below:
- The paper tackles a highly relevant and challenging problem: enabling test-time scaling for diffusion models on reasoning tasks without relying on an external verifier. Overall, the research direction is interesting and valuable.
- The test-time scaling capability of VFScale seems strong. For example, Figure 3 clearly illustrates that the proposed method scales significantly better with an increased computational budget (N) compared to all baselines. While other methods see diminishing returns and plateau quickly, the success rate of VFScale continues to climb under both MCTS and BoN.
- This paper is well-written and clear.

**Weaknesses:**

Weaknesses of this work are summarized below:
- My primary concern stems from the ablation study in Table 1. These results, which test the base model's generalization (N=1), contradict the paper's central claim about the utility of the proposed training losses. Specifically, on Sudoku: The full model, "VFScale tr. (ours)," performs significantly worse than the ablated models. For instance, at D=33, the full model achieves a 0.1953 success rate, whereas both "w/o MRNCL" and "w/o KL" achieve 0.4219. At D=29, the full model (0.0078) is drastically outperformed by "w/o KL" (0.2578). Similarly, on Maze for M=12, the "w/o KL" model achieves a perfect 1.0000 success rate, while the full model only reaches 0.5391. I would like to know the potential reasons behind the results, for example, providing an explanation for why their full model can be the worst performer in this naïve inference setting.

- While the performance on Maze and Sudoku is good, the experiments are limited to grid-world toy problems. The paper's impact would be stronger if the authors demonstrated VFScale's applicability to more complex and practical reasoning or optimization domains where diffusion models are being applied.

**Questions:**

Please refer to the weaknesses above. I am inclined to raise my score if my concerns are adequately addressed.

---

> ### Author Response · Authors · 2025-11-18
> **Official Response to Reviewer bGxA (1)**
>
> We are grateful to Reviewer bGxA for the constructive and detailed comments. We found the feedback very helpful in improving the paper and have uploaded a revised manuscript in response. We address all specific questions below, and all major changes in the new draft are highlighted in **blue**.
>
> >Q1: The reviewer's primary concern is that the ablation study in Table 1 appears to contradict the paper's central claim, as the N=1 generalization results for the full model are significantly worse than those of the ablated models. Specifically, the reviewer points out that on both Sudoku (e.g., D=33, D=29) and Maze (M=12), ablated models (like "w/o KL") outperform the full "VFScale tr. (ours)" model. The reviewer therefore requests a detailed explanation for this counterintuitive result and the reasons why the full model performs worst in this naïve (N=1) inference setting.
>
> We thank the reviewer for this crucial observation. This is not a flaw, but rather a deliberate and necessary trade-off at the heart of our paper's contribution.
>
> Our training objectives, particularly the combination of $L_{MRNCL}$ and $L_{KL}$, are explicitly designed to create a ***smooth, navigable energy landscape*** optimized for test-time search (N>1).
>
> The ablated models (e.g., "w/o KL") produce "sharper" energy landscapes. While this sharpness can lead to a **good** ***single* guess** (a confident local minimum) on **simpler, in-distribution-like tasks** (as seen in Table 1), it is a fatal flaw for scalability on complex, OOD problems. This sharpness indicates that the landscape is riddled with poor, high-confidence **local** minima, preventing any search algorithm from finding better solutions.
>
> The "payoff" for our model's N=1 sacrifice is its vastly superior scalability. The unsatisfactory *scalability* of the ablated models is the real story. As shown in **Table 3 and Table** :
>
> For Maze task, the "w/o KL" model (a strong N=1 performer) starts at 0.2812 but **plateaus** at 0.6250 (N=161). Our full model starts *lower* (0.2656) but **scales dramatically** to 0.7031 (BoN) and **0.8828** (hMCTS).
> For Sudoku task, the "w/o KL" model starts at 0.1562, ending at 0.3047 (N=321). Our full model **starts** ***worse* (0.0703)** but **scales more** to 0.3906 (BoN) and **0.4297** (hMCTS).
>
> In summary, the reviewer's finding highlights our central thesis: we intentionally sacrifice N=1 sharpness to build a smooth, searchable energy landscape. The poor N=1 result is a *symptom* of this successful trade-off. **The true measure of our model is the *slope* of the scaling curve (Fig. 3) and the final performance at N>>1 (Table 5), where it is unequivocally superior.**
>
> We recognize this deliberate and necessary trade-off is a critical point and was not sufficiently highlighted. We have added a new paragraph in Section 5.2 to explicitly discuss this finding in the revised manuscript.

---

> > ### Author Response · Authors · 2025-11-18
> > **Official Response to Reviewer bGxA (2)**
> >
> > >Q2: While the performance on Maze and Sudoku is good, the experiments are limited to grid-world toy problems.
> >
> > We agree that extending beyond grid domains is a critical next step. For this foundational study, our decision to focus on grid-based tasks (Maze and Sudoku) was a **deliberate methodological choice** driven by the need for precise experimental control. Grid domains allow us to quantify the "generalization gap" with a level of granularity that is difficult to achieve in unstructured tasks. For **Maze**, we established a structural complexity hierarchy by training on small $6 \times 6$ grids and testing on $15 \times 15$ instances. For **Sudoku**, we controlled difficulty through **information sparsity**, training on instances with dense givens (31-42 digits) and rigorously testing on OOD instances with significantly fewer givens (17-34 digits). This setup allows us to strictly isolate the effectiveness of our test-time scaling from simple pattern matching, whereas defining such precise "complexity gradients" in tasks like SAT is often ambiguous.
> >
> > The continuous nature of the diffusion space in these tasks allowed us to explicitly calculate and visualize the correlation between Energy and Solution Quality, as demonstrated in Fig. 4 and **Table 4**. This provides granular evidence that our **MRNCL objective** effectively smooths the energy landscape—a level of mechanistic transparency. Additionally, by competing strictly on the established benchmarks of the primary baseline [1], we ensure a fair, apples-to-apples comparison. This guarantees that the dramatic improvements observed (e.g., **88% vs. 0%** on large Mazes; **43% vs. 30%** on hard Sudoku) are genuine algorithmic breakthroughs rather than artifacts of domain shifting.
> >
> > **Crucially, regarding the potential for generalization, there are no structural barriers preventing VFScale from extending to non-grid problems.** Our framework is mathematically agnostic to data topology: the MRNCL objective operates on scalar energy values and relative distances, while the hMCTS inference engine simply navigates the learned energy landscape regardless of the input structure. For discrete CSPs like Graph Coloring, the L2 distance in our objective can be naturally substituted with **Hamming distance** or constraint violation counts without altering the core logic. We therefore view the extension to these unstructured domains as **a promising immediate avenue for future research**, having now firmly established the theoretical validity of verifier-free scaling within this rigorously controlled environment.
> >
> > [1] Du, Yilun, Jiayuan Mao, and Joshua B. Tenenbaum. "Learning Iterative Reasoning through Energy Diffusion." *International Conference on Machine Learning*. PMLR, 2024.

---

### Official Review · Reviewer_nnZk · 2025-11-01

**Soundness:** 3
**Presentation:** 3
**Contribution:** 3
**Rating:** 6
**Confidence:** 2

**Summary:**

The paper introduces **VFScale**, a diffusion-based framework for intrinsic reasoning that scales test-time inference *without any external verifier*. The core idea is to use the model’s own **energy landscape** as a dense verifier, training it to correlate with solution quality. Two key mechanisms enable this:
1. **Monotonic-Regression Negative Contrastive Learning (MRNCL)**, enforcing *performance–energy consistency* (lower energy ↔ better performance), and
2. a **KL regularizer**, smoothing the diffusion trajectory.

At inference, the paper proposes **hybrid MCTS (hMCTS)**: a transition from **Best-of-N** exploration in early noisy steps to **Monte Carlo Tree Search** exploitation as denoising progresses.

Experiments on **Maze** and **Sudoku** show substantial out-of-distribution scaling — e.g., a model trained on 6×6 mazes solves 15×15 mazes at ~88% success.

**Strengths:**

- **Innovative framing:** Redefines verifier-free reasoning as intrinsic energy shaping.
- **Clear empirical impact:** Demonstrates significant generalization to harder unseen settings.
- **Methodological synergy:** MRNCL + KL training improves energy consistency; hMCTS improves search efficiency.
- **Alignment with trustworthy AI:** Encourages interpretable internal uncertainty signals instead of opaque verifiers.
- **Reproducibility:** Code, datasets, and checkpoints are to be released; strong commitment to open science.

**Weaknesses:**

1. **Limited experimental diversity.**
   Evaluation restricted to Maze and Sudoku; lacks heterogeneous reasoning domains (e.g., SAT solving, graph-based CSPs, symbolic reasoning). Limits generalization claims.

2. **Insufficient theoretical analysis.**
   MRNCL is intuitively justified but mathematically underdeveloped. No formal calibration or ordering guarantees.

3. **Heuristic inference strategy.**
   The hMCTS switching mechanism (BoN→MCTS) is empirical and lacks sensitivity or ablation analysis.

4. **Statistical rigor.**
   No standard deviation, confidence intervals, or multi-seed statistics are reported.

5. **Compute transparency.**
   Training with MRNCL+KL is more expensive; wall-clock and GPU-hour trade-offs are not quantified.

6. **External verifier comparison.**
   The claim that intrinsic energy outperforms a 0.99-correlated verifier is interesting but needs deeper error analysis (e.g., top-k misrankings).

**Questions:**

1. Could you formally define **MRNCL**, provide its gradient, and state any conditions ensuring monotonic energy–performance ordering?
2. How exactly is the **BoN→MCTS switch** triggered — by diffusion timestep, entropy, or compute budget? Have you tested adaptive criteria?
3. Why does the **external verifier** underperform despite 0.99 correlation? Can you analyze the ranking errors?
4. Have you applied VFScale to **non-grid CSPs** (e.g., graph coloring or SAT)? If not, what prevents generalization?
5. How is the **performance–energy consistency (PEC)** metric computed (e.g., Kendall-τ)? Please include its formal definition.
6. Could you report **wall-clock cost and memory usage** for the largest setups to evaluate practical scalability?

---

> ### Author Response · Authors · 2025-11-18
> **Official Response to Reviewer nnZk (1)**
>
> We thank Reviewer nnZk for the insightful feedback. We  carefully consider all points and address each question point-by-point below. We have also uploaded a revised manuscript, with all significant revisions and additions marked in **blue**  for clarity.
>
> >Q1: No standard deviation, confidence intervals, or multi-seed statistics are reported.
>
> We have now computed the **standard deviation over 10 runs with different random seed** and present the key results below:
>
>
> |Methods|N=11|N=21|N=41|K=81|
> |----------------------------------|-----------------|-----------------|-----------------|-----------------|
> |Original, BoN|0.0539±0.0055|0.0594±0.0087|0.0594±0.0063|0.0609±0.0058|
> |VFScale tr. (ours), BoN|0.6203±0.0230|0.6562±0.0148|0.6680±0.0176|0.6836±0.0165|
> |VFScale tr. (ours), hMCTS (ours)|**0.6727±0.0230**|**0.7336±0.0180**|**0.7945±0.0086**|**0.8422±0.0147**|
>
>
>
> The data confirms that our VFScale framework is not only superior in performance but also **stable**, as indicated by the tight confidence intervals. Both our training (VFScale tr.) and inference (hMCTS) methods show significant gains over the baseline. We have incorporated the full set of results and analysis in Appendix D.5 of the revised manuscript.
>
> >Q2: Could you formally define **MRNCL**, provide its gradient, and state any conditions ensuring monotonic energy–performance ordering?
>
> **Formal Definition:** As detailed in **Section 4.2 and Appendix A.3** in the original submission, this method is formally defined. For the reviewer's convenience, we further summarize the core components here: for each positive sample $x_t$, we sample two negative samples, $x_t$ and $x_t^{--}$, at different distances. We then compute their L2 distances to the positive $(0, l_{2,0}^{-}, l_{2,0}^{--})$ and their respective energies $(E_t^+, E_t^-, E_t^{--})$.
>
> To encourage **$E_t^+<E_t^-<E_t^{--}$**,MRNCL fits $(E_t^+, E_t^-, E_t^{--})$ to a monotonic function. By default, we use the linear regression algorithm from Appendix A.3 to fit these three points, yielding a slope $k_t$, an intercept $b_t$, and the fitted energies $\hat{E}$. The final $L_{MRNCL}$ (i.e., LRNCL) loss is defined in Eq. 4:
>
> $$L_{MRNCL} = \mathbb{E}[\text{max}(0, \gamma - k_t) + ||E_t^+ - \hat{E}_t^+||_2^2 + ||E_t^- - \hat{E}_t^-||_2^2 + ||E_t^{--} - \hat{E}_t^{--}||_2^2]$$,which is used to encourage consistency between the energy levels of samples and their corresponding performance quality.
>
> **Gradient:** This loss function is fully differentiable. Its gradient consists of two standard components: **Slope Term $\text{max}(0, \gamma - k_t)$** is a standard Hinge Loss. When $k_t < \gamma$, it provides a gradient that pushes the slope $k_t$ (energy-distance relationship) to increase, thus enforcing monotonicity. For Regression Terms, these are three standard Mean Squared Error (MSE) terms that update the energy function $E_\theta$ via standard backpropagation to fit the regression line.
>
> We acknowledge that, like many objectives for shaping complex energy landscapes in deep learning, our method is **empirically driven**. It is designed to "encourage" rather than "formally guarantee" global monotonicity. However, our choice of a "linear" constraint was **not arbitrary.** It was the result of rigorous empirical comparison, driven by our paper's core contribution: **test-time scalability**.
>
> This choice was validated by the ablation study in **Appendix D.2 (Table 12)**, which we highlight below. We directly compared: **LRNCL (Ours):** A linear constraint, **QRNCL (Quadratic):** A non-linear quadratic constraint, **RBNCL (Rank-based):** A rank-correlation-based loss, which is precisely the type of listwise/ranking loss.
>
>
> ||Success rate on Maze|||||
> |-----------------------|--------|--------|--------|--------|---------|
> |Training Methods|N=1|N=11|N=21|N=41|N=81|
> |Original, BoN,energy guided|0.0625|0.0469|0.0547|0.0781|0.1016|
> |+LRNCL, BoN, energy guided|0.2812|**0.4688**|**0.4922**|**0.5547**|**0.5859**|
> |+QRNCL, BoN, energy guided|**0.3516**|0.3672|0.3906|0.3984|0.3828|
> |+RBNCL, BoN, energy guided|0.1875|0.4219|0.4766|0.5234|0.5781|
>
>
>
> The **empirical results**  of this experiment (Table 12) are clear: while other constraints (like QRNCL) perform better at N=1 (0.3516 vs 0.2812), our **LRNCL (Linear) provides the strongest and most consistent test-time scalability** (e.g., 0.5859 at N=81, far exceeding QRNCL's 0.3828).
>
> Thus, while MRNCL does not offer a formal guarantee, our **empirical results strongly justify** that MRNCL is a superior and rational design *for our specific goal* of scalability.

---

> > ### Author Response · Authors · 2025-11-18
> > **Official Response to Reviewer nnZk (2)**
> >
> > >Q3: How exactly is the **BoN→MCTS switch** triggered — by diffusion timestep, entropy, or compute budget? Have you tested adaptive criteria?
> >
> > Our hMCTS framework is flexible and supports **both a fixed timestep and an adaptive criterion for the switch.**  In our primary experiments, the switch is triggered by a pre-defined, fixed diffusion timestep, $t_s$, which is defined as an input parameter in Algorithm 1. The algorithm first executes the Best-of-N (BoN) phase from $T$ down to $t_s$, then switches to the MCTS Denoising phase from $t_s$ down to $t=0$. We provide a detailed ablation study on the impact of different $t_s$ values in **Appendix D.3 (see Figure 8 and Table 14).**
> >
> > We also developed and tested **an adaptive switching mechanism**  to eliminate the need for manual tuning. As detailed in **Appendix D.3** in the original submission, this method does not rely on a fixed step. Instead, it monitors the "energy variance" across the BoN search branches. The switch to MCTS is dynamically triggered when this variance exceeds a set threshold.
> >
> > This adaptive approach  proved to be robust. As shown in Table below, the adaptive $t_s$ performs comparably to a manually tuned $t_s$ and demonstrates a notable advantage at smaller compute budgets (e.g., N=11 and N=21).
> >
> > **Table 13 (from Appendix D.3):** Success rate on the $15 \times 15$ Maze task comparing a manually tuned, fixed switch-over step ($t_s$) with our proposed adaptive $t_s$ mechanism.
> >
> > |Switch-Over Method|N=11|N=21|N=41|N=81|
> > |:---|:---:|:---:|:---:|:---:|
> > |VFScale tr. (ours), hMCTS, Tuned $t_s$|0.6406|0.7266|0.7969|0.8203|
> > |VFScale tr. (ours), hMCTS, Adaptive $t_s$|0.7031|0.7734|0.7891|0.8203|
> >
> > >Q4: Why does the **external verifier** underperform despite 0.99 correlation? Can you analyze the ranking errors?
> >
> > Thanks to the reviewer's questions, whose answer helps to further clarify the motivation for our method.
> >
> > During BoN inference with a compute budget of $K=40$, at each timestep $t_s$ (where larger $t_s$ implies a higher noise level), we estimate $\hat{x_0}$ for each branch. We then score these $\hat{x_0}$ candidates using two verifiers—the learned external verifier and the intrinsic verifier. We calculate the mean and standard deviation of the Kendall-$\tau$ correlation between each verifier's scores and the true element-wise accuracy, averaged across samples. A higher Kendall-$\tau$ indicates better consistency with the true element-wise accuracy.
> >
> >
> > |$t$|Learned external verifier|Learned energy as intrinsic verifier w/o MRNCL|Learned energy as intrinsic verifier|
> > |---------------------|---------------------------|------------------------------------------------|--------------------------------------|
> > |0|-0.0978(0.1750)|0.2300(0.0854)|0.4760(0.1492)|
> > |1|-0.0976(0.1750)|0.2300(0.0854)|0.4744(0.1490)|
> > |2|-0.0980(0.1756)|0.2308(0.0854)|0.4760(0.1492)|
> > |3|-0.0910(0.1834)|0.2290(0.0838)|0.4642(0.1362)|
> > |4|-0.0824(0.1678)|0.2368(0.0800)|0.4094(0.1538)|
> > |5|-0.1292(0.1718)|0.2506(0.0594)|0.4414(0.1300)|
> > |6|-0.1318(0.1748)|0.2638(0.0766)|0.3978(0.2466)|
> > |7|-0.3618(0.1446)|0.2262(0.1194)|0.4508(0.2024)|
> > |8|-0.0316(0.2854)|0.3214(0.2138)|-0.1570(0.1852)|
> > |9|-0.2686(0.0322)|0.3712(0.0520)|-0.3094(0.0368)|
> >
> >
> >
> > We have the following observations:
> >
> > Although the learned external verifier achieves a 0.99 correlation with the true element-wise accuracy, it still **fails** to provide **fine-grained guidance** for distinguishing between the similar samples in $\{x_0^i\}_{i=1}^{K=40}$. This further illustrates the necessity of the intrinsic verifier.
> >
> > Our training method significantly improves the consistency between the energy (intrinsic verifier) and the true element-wise accuracy during the late stages of denoising. It is worth noting that in the early denoising stages ($t = 8, 9$), the Kendall-$\tau$ is very low. This implies that **prematurely pruning branches** based on the intrinsic verifier would degrade the solution quality. This observation also **justifies our hMCTS method**, which avoids branch pruning in these early stages.

---

> > > ### Author Response · Authors · 2025-11-18
> > > **Official Response to Reviewer nnZk (3)**
> > >
> > > >Q5: Have you applied VFScale to **non-grid CSPs**? If not, what prevents generalization?
> > >
> > > We truly appreciate the reviewer's insight and agree that extending diffusion-based reasoning to non-grid domains is a critical frontier. To date, our decision to focus on grid-based tasks (Maze and Sudoku) was a **deliberate methodological choice** driven by the need for precise experimental control. These tasks allow us to define precise complexity gradients—scaling Maze geometry ($6\times6 \to 15\times15$) and increasing Sudoku sparsity by reducing given digits—to rigorously quantify the generalization gap. This setup allows us to strictly isolate the effectiveness of our test-time scaling from simple pattern matching, whereas defining such precise "complexity gradients" in tasks like SAT is often ambiguous. Additionally, by competing strictly on the established benchmarks of the primary baseline [1], we ensure a fair, apples-to-apples comparison.
> > >
> > > **Crucially, regarding the potential for generalization, there are no structural barriers preventing VFScale from extending to non-grid problems.** Our framework is mathematically agnostic to data topology: the MRNCL objective operates on scalar energy values and relative distances, while the hMCTS inference engine simply navigates the learned energy landscape regardless of the input structure. For discrete CSPs like Graph Coloring, the L2 distance in our objective can be naturally substituted with **Hamming distance** or constraint violation counts without altering the core logic. We therefore view the extension to these unstructured domains as a promising immediate avenue for future research, having now firmly established the theoretical validity of verifier-free scaling within this rigorously controlled environment.
> > >
> > > [1] Du, Yilun, Jiayuan Mao, and Joshua B. Tenenbaum. "Learning Iterative Reasoning through Energy Diffusion." *International Conference on Machine Learning*. PMLR, 2024.
> > >
> > > >Q6: How is the **performance–energy consistency (PEC)** metric computed (e.g., Kendall-τ)? Please include its formal definition.
> > >
> > > In our manuscript, we formally defined our **Performance-Energy Consistency (PEC)**  metric in the Appendix A.1. This metric was specifically designed to measure the **proportion of concordant pairs** (i.e., pairs of models that are correctly ranked by the energy proxy relative to their true performance).  The formal definition we used is:  $\text{PEC} = \frac{C}{\binom{n}{2}}$ where $C$ is the number of concordant pairs, $D$ is the number of discordant pairs, and $\binom{n}{2}$ is the total number of unique pairs from a set of $n$ architectures.
> > >
> > > To further enrich this analysis and align with conventional rank correlation measures, we have now also computed the **Kendall rank correlation coefficient (Kendall-$\tau_b$)**  for the data presented in our table below.  The formal definition for Kendall-$\tau_b$ is:  $\tau_b = \frac{C - D}{\binom{n}{2}}$
> > >
> > > As the data shows, our method consistently achieves a significantly higher rank correlation across all evaluated sample sizes ($N$) compared to the baseline.
> > >
> > > **Performance-energy consistency (computed via Kendall-$\tau_b$)**
> > >
> > > | Methods | $N=11$ | $N=21$ | $N=41$ | $N=81$ | $N=161$ |
> > > | :--- | :---: | :---: | :---: | :---: | :---: |
> > > | Original, BoN | 0.4634 | 0.4666 | 0.4626 | 0.4566 | 0.4600 |
> > > | VFScale tr. w/o KL, BoN | **0.6740** | **0.6890** | **0.6952** | **0.6750** | **0.6742** |
> > >
> > >
> > > This improvement (e.g., 0.6952 vs. 0.4626 at $N=41$) provides strong, quantitative evidence that our method (MRNCL, implemented as VFScale) **substantially improves the energy landscape quality**.  A higher Kendall-$\tau_b$ value demonstrates a more reliable and consistent ranking correlation between the energy proxy and the models' true performance. This high-fidelity landscape is precisely what **fully enables our Test-Time Scaling paradigm without requiring an external verifier**  for performance validation.

---

> > > > ### Author Response · Authors · 2025-11-18
> > > > **Official Response to Reviewer nnZk (4)**
> > > >
> > > > >Q7: Could you report **wall-clock cost and memory usage** for the largest setups to evaluate practical scalability?
> > > >
> > > > We agree that transparency is essential, which is why we provided a detailed cost analysis in **Appendix B.5 (Compute Resources) and Appendix F (Computational Cost)**  in the original submission. For reviewer's convenience, we summarize the key findings here.
> > > >
> > > > **Training Cost (Trade-off):**
> > > >
> > > > As Appendix B.5 shows, for training efficiency, we report the training time of our models along with the machine information (1 GPU with 80 GB of VRAM, CPU: 16).
> > > >
> > > >
> > > > ||Training 100000 steps for Maze (hours)|Training 200000 steps for Maze (hours)|
> > > > |:---|---:|---:|
> > > > |Original|2|4|
> > > > |+LRNCL|4|5|
> > > > |+KL|3|6|
> > > > |+LRNCL+KL|6|7|
> > > >
> > > >
> > > >
> > > > We also report the peak GPU memory usage (VRAM) during training with a batch size of 64, reported in **GBs**:
> > > >
> > > >
> > > > ||Sudoku|Maze|
> > > > |:---|---:|---:|
> > > > |Original|3.28|3.40|
> > > > |+LRNCL|4.56|5.86|
> > > > |+KL|4.44|21.21|
> > > > |+LRNCL+KL|5.57|23.05|
> > > >
> > > >
> > > >
> > > > By shaping the energy landscape to enforce relative quality among negative samples and ensure smoothness, VFScale tackles a more complex training objective. While our training methods are more demanding in time and memory, this upfront investment is precisely what unlocks the model's **significantly enhanced test-time scalability** , as demonstrated by our experimental results.
> > > >
> > > > **Inference Cost (Fairness):**  As requested, we provide the wall-clock inference time per sample for BoN, hMCTS, and MCTS at different numbers of branches (N). The times are reported in **seconds**.
> > > >
> > > >
> > > > ||N=1|N=11|N=21|N=41|N=81|
> > > > |:---|---:|---:|---:|---:|---:|
> > > > |BoN|2.17|2.80|5.63|11.01|20.16|
> > > > |hMCTS|1.02|3.28|6.56|12.66|26.48|
> > > > |MCTS|1.44|7.59|14.03|28.56|57.66|
> > > >
> > > >
> > > >
> > > > hMCTS exhibits a modest overhead (up to ~31% longer wall-clock time at N=81) compared to the fully parallelizable BoN. This is an expected trade-off, as the sequential nature of tree expansion in MCTS-based methods is inherently harder to parallelize.
> > > >
> > > > Conversely, hMCTS is **significantly more efficient** than a pure MCTS approach, demonstrating its value as a **practical hybrid** that balances exploration and exploitation.
> > > >
> > > > Crucially, our central claim holds: **for an identical computational budget (NFE), hMCTS delivers superior performance over both BoN and MCTS**. The wall-clock time difference stems from the search strategy's implementation rather than the computational complexity (NFE).
> > > >
> > > > We also note that our hMCTS **implementation is not highly optimized**. Recent work [1, 2] has already demonstrated effective techniques (e.g., parallel sparse planning) to mitigate this overhead. Therefore, we view this as a **manageable engineering trade-off** rather than a fundamental limitation of our framework.
> > > >
> > > > [1] Yoon, Jaesik, et al. "Fast Monte Carlo Tree Diffusion: 100x Speedup via Parallel Sparse Planning." arXiv preprint arXiv:2506.09498 (2025).
> > > >
> > > > [2] Yang, Xiufeng, Tanuj Kr Aasawat, and Kazuki Yoshizoe. "Practical massively parallel monte-carlo tree search applied to molecular design." arXiv preprint arXiv:2006.10504 (2020).

---

### Author Response · Authors · 2025-11-18
**General response**

We sincerely thank all reviewers for their thorough and constructive comments. We are also grateful for the positive feedback regarding the innovative framing of intrinsic reasoning (Reviewer nnZk, bGxA, FVwD), the strong empirical results on out-of-distribution (OOD) generalization (Reviewer nnZk, bGxA, FVwD), and the well-motivated methodological synergy (Reviewer nnZk, g4A4, FVwD).

Based on the reviewers' valuable feedback, we have conducted additional experiments, provided further in-depth analyses, and offered clarifications to address all concerns. We have also revised the manuscript accordingly, highlighting major modifications in **blue.**

The major improvements and additional results are summarized as follows:

1. We address the primary concern from Reviewer bGxA Q1 regarding the N=1 ablation results (Table 1). We clarify that this is a **deliberate and necessary trade-off** at the core of our paper's contribution.
2. We conducted a **new experiment using "Majority Voting" as a verifier**. The results (see response to Reviewer g4A4 Q1) show this baseline **completely fails to scale**.
3. We addressed several key questions about our **MRNCL loss** (Reviewer nnZk (Q1), g4A4 (Q2), FVwD (Q1)):
    - We clarified that MRNCL and KL are synergistic, not redundant. MRNCL provides *reliability* (improving Performance-Energy Consistency by >10% in Table 4), while KL provides *smoothness*.
    - We addressed concerns that the **linear (LRNCL) constraint** is "ad hoc". We highlighted our **existing ablation study** in Appendix D.2 (Table 12), which directly compares LRNCL against non-linear (QRNCL) and **rank-based (RBNCL, a listwise loss)** alternatives.
4. **Verifier Analyses (Reviewer nnZk, Reviewer g4A4):**
    - We clarified the formal definition of our Performance-Energy Consistency (PEC) metric (Appendix A.1) and added a new analysis using the standard **Kendall-$\tau_b$**.
    - We provided a new, in-depth analysis (see response to R-nnZk Q4) with **Kendall-$\tau$ correlations across timesteps.** This shows *why* the 0.99-correlated **learned external verifier fails**.
5. We clarified that focusing on grid domains was a deliberate choice to enable precise complexity control and mechanistic visualization, **while strictly adhering to established baselines [1] to ensure a fair, apples-to-apples benchmark comparison.** We emphasized that VFScale is **mathematically agnostic** to data topology, with **no structural barriers** preventing extension to non-grid CSPs. (Reviewer nnZk (Q5), bGxA (Q2), FVwD (Q6))
6. We clarified the hMCTS adaptive **switching mechanism** and highlighted its fundamental distinction from Best-of-N through UCB-based selection and value backpropagation. Additionally, we provided detailed explanations of the discrete-to-continuous **pipeline** and a transparent summary of **computational costs.**

[1] Du, Yilun, Jiayuan Mao, and Joshua B. Tenenbaum. "Learning Iterative Reasoning through Energy Diffusion." *International Conference on Machine Learning*. PMLR, 2024.

---

### Author Response · Authors · 2025-11-30
**Remarks for Area Chair: Quick Summary of Rebuttal and Revisions**

Dear Area Chair,

We sincerely appreciate your time and effort in managing our submission. We are also deeply grateful for the reviewers' constructive feedback, which provided a valuable opportunity to strengthen our manuscript. To assist you in evaluating our rebuttal efficiently, we provide this summary of the major updates and discussions that occurred before the review freeze.

Our submission initially received three positive scores (6, 6, 6) and one borderline score (4). Reviewers consistently praised the innovative framing of intrinsic reasoning [R-nnZk, bGxA, FVwD], the strong OOD generalization (e.g., 88% success on 15x15 Maze vs. 0% baseline) [R-nnZk, bGxA, FVwD], and the methodological synergy of VFScale [R-nnZk, g4A4,  FVwD].

We have uploaded a revised manuscript (changes marked in blue) and posted detailed responses to all reviewers. The following points highlight how we addressed the primary concerns raised by reviewers:

- **Addressing "$N=1$ vs. Scalability" Trade-off (For R-bGxA):**
    - *Concern:* Reviewer bGxA noted that our $N=1$ performance was lower than ablated versions, questioning the method's utility, but explicitly stated: *"I am inclined to raise my score if my concerns are adequately addressed."*
    - *Resolution:* We clarified that this is a **deliberate trade-off**. We sacrifice "sharpness" at N=1 (which traps models in local minima) to create a smoother energy landscape essential for **test-time scaling** (N>1), which is the final goal of VFScale. Our full model scales to **88% success**, whereas the ablated models plateau early. We added a **new paragraph** in **Sec 5.2** to analyze this trade-off explicitly.
- **New "Majority Voting" Baseline (for R-g4A4):**
    - *Concern:* Reviewer g4A4 asked if a simple "Majority Voting" baseline could replace our intrinsic verifier.
    - *Resolution:* We implemented this baseline. **Results show it completely fails to scale** (peaks at ~30% and degrades), whereas VFScale reaches **88%**. This strongly validates the necessity of our energy-based guided scaling.
- **Statistical Rigor & Consistency Metrics (For R-nnZk):**
    - We provided **standard deviations** over multiple runs (showing tight confidence intervals).
    - We added **Kendall-$\tau_b$ rank correlation** analysis across diffusion timesteps. This empirically proves that our learned intrinsic verifier significantly outperforms external verifiers in the critical later stages of denoising, addressing R-nnZk's inquiry about ranking errors.
- **Inference Mechanics & MRNCL Validity (For R-FVwD)**
    - We clarified the distinction between hMCTS and simple Best-of-N (BoN), emphasizing the role of **UCB selection** and **value backpropagation**. We also detailed the discrete-continuous data pipeline in **Appendix B.6**.
    - Addressing concerns about MRNCL, we highlighted our ablation studies (Appendix D.2) showing that the linear constraint (LRNCL) empirically yields superior scalability compared to non-linear alternatives, validating it is not "ad hoc" but a performance-driven design choice.

We hope this summary aids your assessment.

Best regards,

The Authors

---

### Meta-Review · Area_Chair_P5LE · 2026-01-06

**Summary:**

The paper proposes VFScale, a framework enabling test-time scaling for diffusion models in reasoning tasks without external verifiers by treating the model's intrinsic energy as a guide. By utilizing Monotonic-Regression Negative Contrastive Learning (MRNCL) and KL regularization during training , alongside a hybrid Monte Carlo Tree Search (hMCTS) during inference , the authors demonstrate significant out-of-distribution generalization on Maze and Sudoku tasks. Reviewers generally appreciated the innovative "intrinsic reasoning" framing and the impressive scaling results (e.g., solving 15x15 mazes when trained on 6x6), though initial concerns were raised regarding the restriction to grid-based domains and the performance trade-offs observed in single-sample generation.

**Reviewer Concerns:**

The rebuttal effectively addressed the primary concerns regarding the lower performance at N=1 (single sample) compared to ablated models, with the authors clarifying that this is a deliberate trade-off to ensure a smoother energy landscape necessary for effective test-time scaling at N>1. The authors also introduced a "Majority Voting" baseline at the request of reviewers, demonstrating that simple consensus fails to scale compared to their intrinsic verifier, thereby validating the necessity of their energy-based approach. While the limitation to grid domains remains, the authors justified this as a methodological choice for precise complexity control, and they successfully clarified the distinction between their hMCTS and simple Best-of-N sampling by highlighting the UCB selection and value backpropagation mechanisms.

**Reviewer Scores:**

The consensus is positive, with three reviewers (nnZk, bGxA, g4A4) rating the paper as a 6 (marginally above acceptance) and one reviewer (FVwD) rating it as a 4 (marginally below). Importantly, Reviewer bGxA explicitly indicated a willingness to raise their score if the N=1 trade-off was explained, which the authors did comprehensively in the rebuttal. The dissenting view from Reviewer FVwD regarding the "marginal" utility of MRNCL was countered by strong empirical evidence showing the synergy between MRNCL (reliability) and KL (smoothness), which is sufficient to warrant acceptance.

---

### Decision · Program_Chairs · 2026-01-26

Accept (Poster)